# Partitioning of fatty acids between membrane and storage lipids controls ER membrane expansion

Pawel K Lysyganicz [ID][1], Antonio D Barbosa [ID][1], Shoily Khondker[2], Nicolas A Stewart[3], George M Carman [ID][2], Phillip J Stansfeld [ID][4], Marcus K Dymond [ID][3] & Symeon Siniossoglou [ID][1✉]

## Abstract

**Biogenesis of membrane-bound organelles involves the synthesis, remodeling, and degradation of their constituent phospholipids. How these pathways regulate organelle size remains poorly understood. Here we demonstrate that a lipid-degradation pathway inhibits expansion of the endoplasmic reticulum (ER) membrane. Phospholipid diacylglycerol acyltransferases (PDATs) use endogenous phospholipids as fatty-acyl donors to generate triglyceride stored in lipid droplets. The significance of this non-canonical triglyceride biosynthesis pathway has remained elusive. We find that the activity of the yeast PDAT Lro1 is regulated by a membrane-proximal helical segment facing the luminal side of the ER bilayer. To reveal the biological roles of PDATs, we engineered an Lro1 variant with derepressed activity. We show that active Lro1 mediates retraction of ER membrane expansion driven by phospholipid synthesis. Furthermore, subcellular distribution and membrane turnover activity of Lro1 are controlled by diacylglycerol produced by the activity of Pah1, a conserved member of the lipin family. Collectively, our findings reveal a lipid-metabolic network that regulates endoplasmic reticulum biogenesis by converting phospholipids into storage lipids.**

**Keywords** Phospholipid; Lipid Droplet; Membrane; Endoplasmic Reticulum; Yeast
**Subject Category** Organelles

## Introduction

Eukaryotic cells store fatty acids (FAs) in the form of triglycerides (TGs). Because of their non-polar nature, TGs and other "neutral" lipids such as steryl esters, are packed into specialized organelles called lipid droplets (LDs), which are generated from, and associate with, the endoplasmic reticulum (ER) membrane (Olzmann and Carvalho, 2019). The ability to store TGs in LDs is critical for key physiological functions of eukaryotic cells. Fatty acids that are

mobilized from TGs provide metabolic energy when nutrients are scarce, as well as lipid intermediates for the biogenesis of membranes to support cell growth. TGs also provide a metabolic "sink" that protects cells against the lipotoxicity of excess free fatty acids. Disruption of balanced TG storage is responsible for widespread pathologies of the modern world including type 2 diabetes, non-alcoholic fatty liver disease and obesity (Krahmer et al, 2013).

Because of its significance, the pathway of TG synthesis is universally conserved in eukaryotic kingdoms. TG is made by the esterification of acyl-CoA-activated FAs to a diacylglcerol (DG) backbone, and catalyzed by diacylglycerol O-acyltransferases (DGATs) (Chen et al, 2022). Fungi, green algae and plants can generate TG via an alternative mechanism which, so far, has received little attention. In this acyl-CoA-independent pathway, cells use endogenous membrane lipids as FA donors to esterify DG. This reaction is catalyzed by phospholipid diacylglycerol acyltransferases (PDATs) and yields TG and a lysophospholipid (Fig. 1A) (Dahlqvist et al, 2000; Oelkers et al, 2000). Although mammals do not express any apparent PDAT, an analogous pathway was recently proposed to operate in humans (McLelland et al, 2023). In budding yeast, the PDAT Lro1 deacylates two abundant membrane phospholipids, phosphatidylethanolamine (PE) and phosphatidylcholine (PC) (Dahlqvist et al, 2000; Oelkers et al, 2000), and partitions dynamically between the ER and an inner nuclear membrane (INM) subdomain in a cell-cycle-dependent manner (Barbosa et al, 2019). This membrane subdomain has the unique property of expanding in response to increased phospholipid levels, while the rest of the nuclear envelope does not (Witkin et al, 2012). Based on these observations, it was proposed that the activity of Lro1 regulates nuclear membrane expansion, a process which normally takes place during mitosis in yeast cells. Notably, while TG accumulation via the canonical, acyl-CoA-dependent, pathway is a response to stress or nutrient limitation, Lro1 is more active in making TG during favorable, nutrient-replete conditions (Oelkers et al, 2002). A similar situation has been described in the green alga *Chlamydomonas reinhardii*, where its chloroplast-associated PDAT is more active during the exponential phase of growth (Yoon et al, 2012). Hence, how PDAT activity relates to the established metabolic roles of TG remains unknown. Given that their substrates are phospholipids, we considered the possibility that PDATs respond to the need of organelle membrane reorganization.

[1]Cambridge Institute for Medical Research, University of Cambridge, Cambridge CB2 0XY, UK. [2]Department of Food Science and the Rutgers Center for Lipid Research, Rutgers University, New Brunswick, NJ 08901, USA. [3]Centre for Lifelong Health, University of Brighton, Brighton, UK. [4]School of Life Sciences and Department of Chemistry, University of Warwick, Coventry, UK. ✉E-mail: ss560@cam.ac.uk

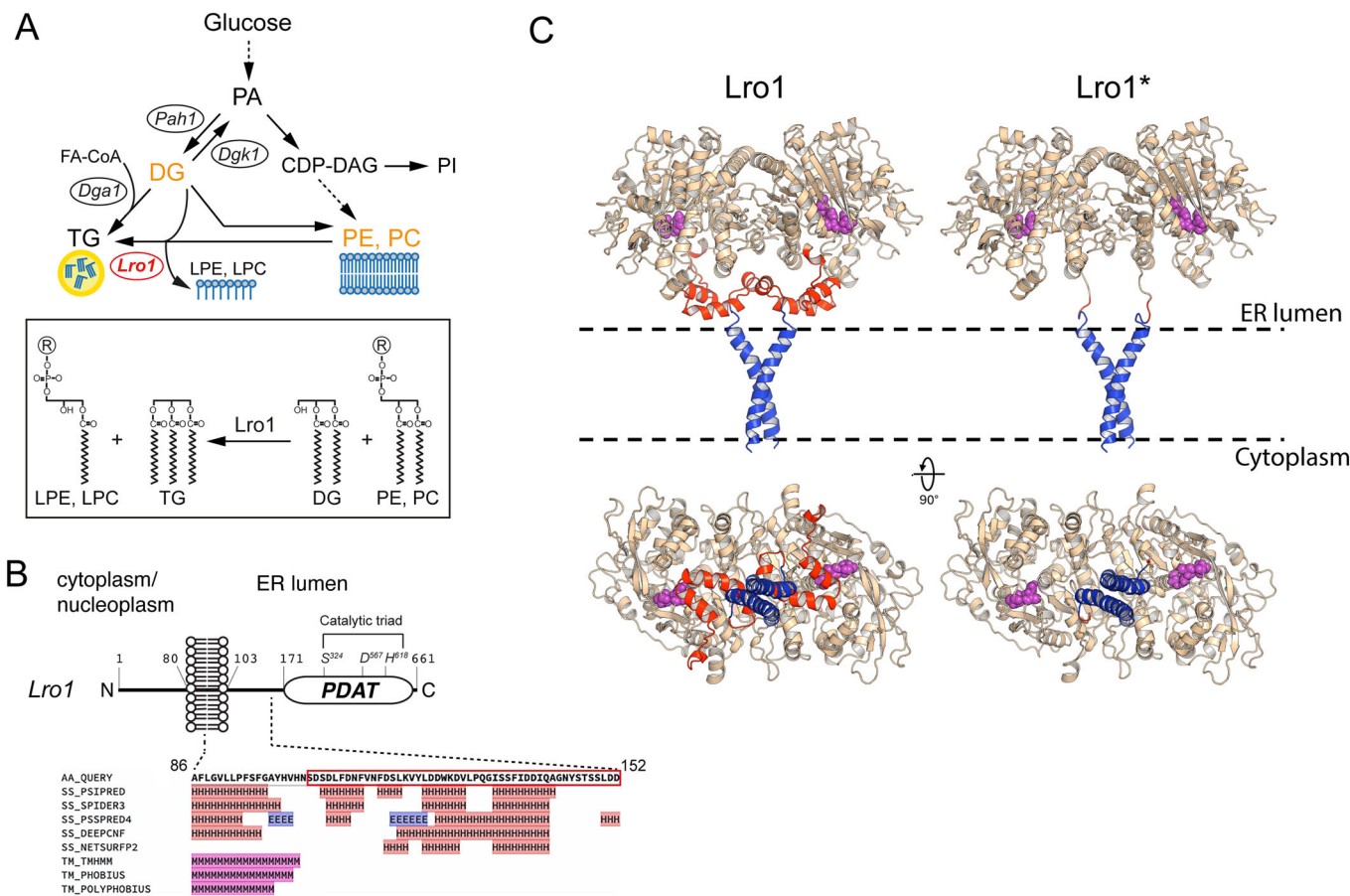

**Figure 1. A predicted membrane-proximal helical segment in the yeast PDAT Lro1.**

(A) Schematic depicting the major lipid metabolic pathways in yeast and the activity of Lro1; PA phosphatidate, DG diacylglycerol, TG triacylglycerol, FA-CoA fatty-acyl-CoA, LPE lysophosphatidylethanolamine, LPC lysophosphatidylcholine, PE phosphatidylethanolamine, PC phosphatidylcholine, PI phosphatidylinositol. (B) Schematic of the topology of Lro1 with respect to the nuclear/ER membrane. The catalytic triad within the PDAT domain is indicated. The various secondary structure prediction results for the sequence that links the transmembrane helix to the PDAT domain are shown. The boxed sequence denotes residues Ser104 to Asp152. H, alpha helix; E, beta strand; M, transmembrane domain. Data are from the HHpred Quick2D tool (Zimmermann et al, 2018). (C) Models of wild-type Lro1 (left) and Lro1* (right). The Ser104–Asp152 domain is colored red and the residues of the active site are shown as purple spheres. The bottom panels depict the same models rotated by 90 degrees. The transmembrane helices are colored blue. The dashed lines indicate the position of the lipid bilayer. The models were built from residue R71 to M661.

To address these questions, we established a genetic system to manipulate in vivo the PDAT activity of Lro1. We found that the PDAT activity of Lro1 is inhibited by a predicted helical segment that faces the luminal side of ER membrane. By using a genetically engineered Lro1, which lacks this sequence and is catalytically derepressed, we show that Lro1 can retract the expansion of phospholipid-driven ER membranes. Moreover, we find that DG, provided by the lipin Pah1, is the major factor that controls the cellular distribution and PDAT activity of Lro1. Our data establish that cells regulate membrane-bound organelle biogenesis by mobilizing fatty acyl chains from phospholipids to TGs.

## Results

### Modeling of the protein structure of Lro1

To gain insights into the function of PDATs, we sought to generate a system to manipulate their catalytic activity in vivo. We noticed that despite strong overexpression of Lro1, cells increased only moderately their TG levels with no significant effect on their growth rate or ER organization ((Barbosa et al, 2019); our data in this study). Therefore, we hypothesized that the activity of Lro1 is subject to negative regulation in vivo. To explore this possibility, we examined the predicted organization of Lro1. Like all PDATs, Lro1 exhibits a type 2 membrane protein topology with an amino-terminal cytoplasmic sequence, a single transmembrane domain, and a larger catalytic domain, which is glycosylated and located in the ER/perinuclear lumen (Choudhary et al, 2011; Wang and Lee, 2012). The catalytic domain is connected to the transmembrane helix through a "linker" sequence of ~70 residues (Fig. 1B). While most fungal PDATs contain linkers of similar amino acid length, in plant enzymes a significant part of the linker segment is missing (Appendix Fig. S1).

Several secondary structure prediction algorithms suggest the presence of at least two helical segments within the linker segment of Lro1 (Gabler et al, 2020; Zimmermann et al, 2018) (Fig. 1B). Modeling of the protein structure of Lro1, using AlphaFold2

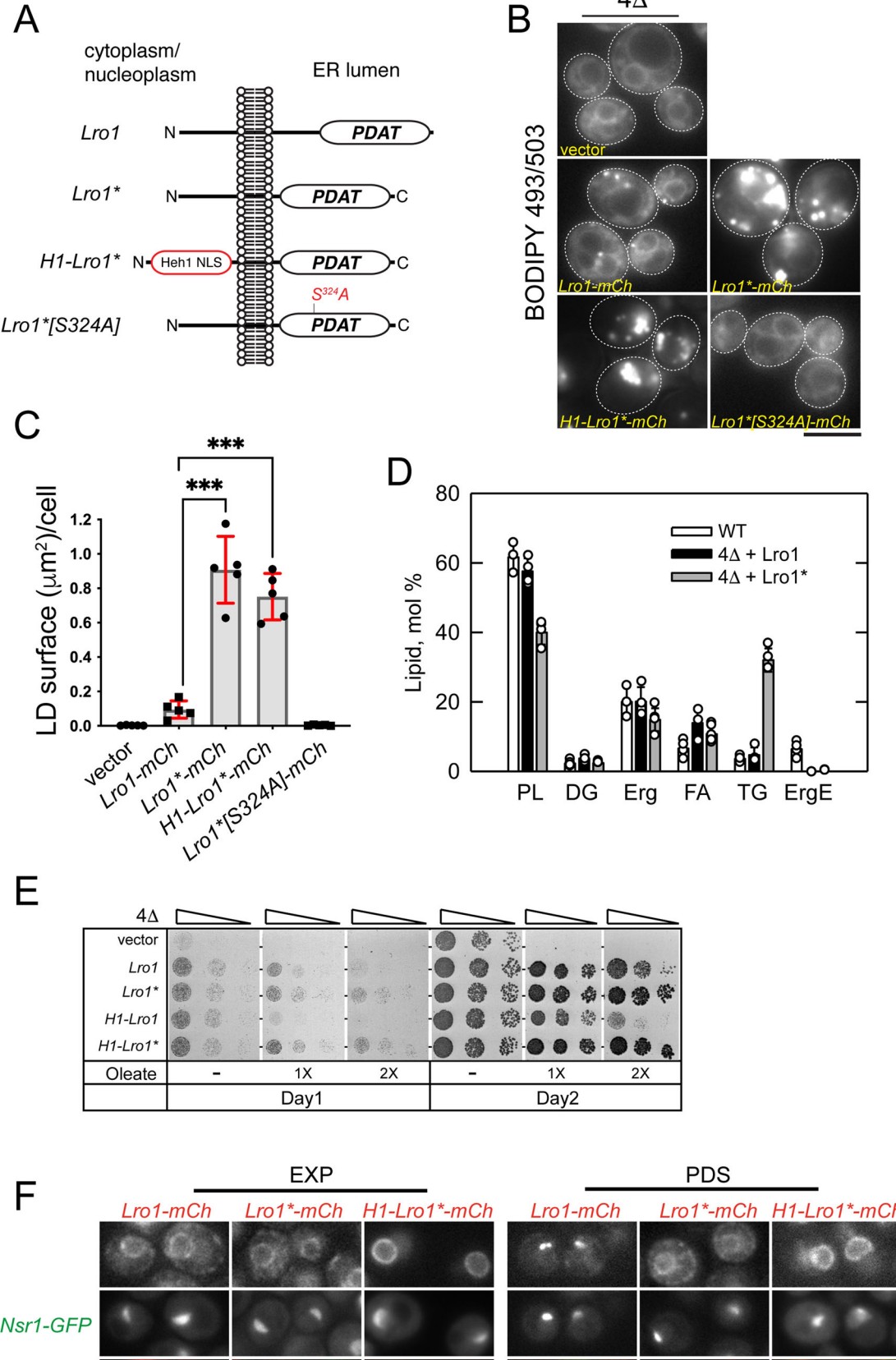

**Figure 2.  Removal of a luminal helical segment generates a hyperactive Lro1 enzyme.**

(**A**) Schematic of Lro1 mutants and their topology with respect to the nuclear/ER membrane. The INM targeting sequence (Heh1-NLS) is indicated. (**B**) Exponentially growing 4Δ cells carrying an empty vector or the indicated Lro1 constructs were stained with BODIPY 493/503 to label LDs; the dotted lines indicate the cell contours. (**C**) Quantification of the BODIPY 493/503 labeling shown in (**B**); data are shown as means ± SD from five experiments with at least 140 cells measured per strain per experiment. Statistical analysis was performed by Forsythe and Welch ANOVA test with Dunnett's T3 correction; ***$P < 0.001$ (Lro1-mCh vs Lro1*-mCh $P = 0.001$; Lro1-mCh vs H1-Lro1*-mCh $P = 0.0005$). (**D**) Lipid composition of the indicated strains in the exponential phase of growth. Cells were grown in the presence of [2-$^{14}$C]acetate, and lipids were extracted, separated, and quantified by one-dimensional TLC as described in Methods; each data point represents the mean ± SD from at least three experiments. DG diacylglycerol, Erg ergosterol, ErgE ergosterol ester, FA fatty acid, PL phospholipid, TG triacylglycerol. (**E**) Cells were grown in selective media and spotted serially on YEPD plates in the absence or presence of two different concentrations of oleate. Plates were grown at 30 °C and scanned after one or 2 days. (**F**) *lro1Δ* cells expressing a chromosomally integrated nucleolar reporter (Nsr1-GFP) and the denoted Lro1-mCh proteins were visualized at the exponential (EXP) or PDS phases of growth. Scale bars in all micrographs, 5 μm. Source data are available online for this figure.

(Jumper et al, 2021), predicts that it forms a dimer. This is consistent with the presence of a GxxxG dimerization motif within its transmembrane domain, which is highly conserved within all fungal Lro1 orthologues (Appendix Fig. S2). Notably, the model depicts the linker sequence forming two short helices with their axes parallel to the membrane surface. We noticed that the helices are positioned so as to block access to the catalytic triad of Lro1 from the membrane surface (Fig. 1C, Lro1 bottom view). Therefore, we postulated that the membrane-proximal helices control the accessibility of the lipid substrate to the catalytic site of Lro1. The model of Lro1 lacking the residues Ser104–Asp152, which make up the helical linker, and thereafter called Lro1*, shows that the catalytic triad is now freely accessible from the membrane surface (Fig. 1C, Lro1* bottom view). Accordingly, we hypothesized that Lro1* exhibits elevated PDAT activity and set up an experimental plan to test this prediction in vivo.

## Removal of the luminal linker of Lro1 derepresses its PDAT activity

To determine how the removal of the luminal helices influences the activity of Lro1, we used a yeast strain (4Δ) lacking the two DG acyltransferases (Lro1 and Dga1) and the two steryl acyltransferases (Are1 and Are2), which is devoid of LDs (Jacquier et al, 2011; Oelkers et al, 2002). This setup allowed us to determine the TG and LDs that are derived specifically from the activity of Lro1. Expression of mCherry-tagged wild-type Lro1 in exponentially growing cells (Lro1-mCh) led to the appearance of small LDs, consistent with the fact that yeast cells do not normally accumulate neutral lipids during this growth phase. On the other hand, the expression of Lro1*-mCh led to the formation of much larger LDs, visualized by BODIPY 493/503 labeling (Fig. 2A–C). Untagged Lro1* had the same effect on LDs (Appendix Fig. S3). We confirmed the presence of larger LDs by co-expressing a neon-green-tagged LD-associated protein (Appendix Fig. S4). This increase in LD size was accompanied by significantly higher TG levels and a reciprocal decrease in cellular phospholipid levels (Fig. 2D), consistent with an increase in the PDAT activity of Lro1*. Protein levels of Lro1* were similar to those of Lro1 (Appendix Fig. S5); moreover, mutation of Ser324 in the lipase motif of Lro1*, which is essential for its catalytic activity, abolished the formation of LDs (Fig. 2B,C). Therefore, the formation of large LDs is mediated by the PDAT activity of Lro1*.

Next, we used a different system to assay the activity of Lro1*. The 4Δ strain lacks the capacity to store excess free fatty acids into LDs and therefore loses viability in the presence of oleate

(Petschnigg et al, 2009). Expression of Lro1*, led to better growth of 4Δ when compared to Lro1, especially at higher oleate concentrations (Fig. 2E; Appendix Fig. S6A). Interestingly, despite the significant decrease in phospholipid levels, Lro1*-expressing cells did not show any growth defect in standard-rich media (Fig. 2E).

When overexpressed from the galactose (Gal) promoter, Lro1 resulted in only a twofold increase of LD size (Fig. 3A). Gal-Lro1* induced a sevenfold increase with cells containing strikingly larger LDs, visualized both by BODIPY labeling or a LD-associated protein (Fig. 3B), consistent with the inhibitory role of the luminal helices. Growth was strongly inhibited in Gal-Lro1*-expressing cells in a manner that depended on its catalytic activity (Fig. 3C, compare Lro1* with Lro1*S324A; Appendix Fig. S6B). We used mass spectrometry to characterize the lipidome of these cells. Gal-Lro1* cells contained an eightfold, excess of TG (Fig. 3D); levels of DG were lower, while lysoPC and lysoPE were higher (Fig. 3D). Notably, levels of the known phospholipid substrates of Lro1—PC and PE—did not change in response to elevated PDAT activity (Fig. 3D); instead, we found significant changes in the levels of the four most abundant PC molecular species consistent with fatty-acyl remodeling of PC taking place in Gal-Lro1* cells (Fig. 3D). PE species showed less evidence of remodeling (Fig. EV1A). Interestingly, and in contrast to PC and PE, total PI levels exhibited a 50% decrease which was detected across all PI species (Figs. 3D and EV1B). Collectively, the changes in the lipidome of Gal-Lro1* cells are consistent with elevated PDAT activity; moreover, our data indicate that cells have mechanisms to maintain the levels of PC and PE despite the re-routing of their fatty acyl chains to TG.

Next, we examined the subcellular distribution of Lro1*. Wild-type Lro1 partitions between the ER and the inner nuclear membrane domain that associates with the nucleolus in a cell cycle-dependent manner (Barbosa et al, 2019). Lro1*, however, localized predominantly to the ER (Fig. 2F). We wondered whether the elevated activity of Lro1* is due to its retention to the ER, where the machinery responsible for TG synthesis and packing into LDs predominantly resides. This is not the case because constitutive redirection of Lro1* to the INM by fusing it to the INM targeting sequence of Heh1 (H1-Lro1*; Fig. 2F) (Meinema et al, 2011) still resulted in large LDs when expressed in the 4Δ strain (Fig. 2A–C). Moreover, H1-Lro1* rescued more efficiently than Lro1 or H1-Lro1 the lipotoxicity of 4Δ (Fig. 2E). Therefore, the elevated activity of Lro1* is independent of its cellular location. Taken together, these data identify a critical sequence within the luminal portion of Lro1 which inhibits its PDAT activity.

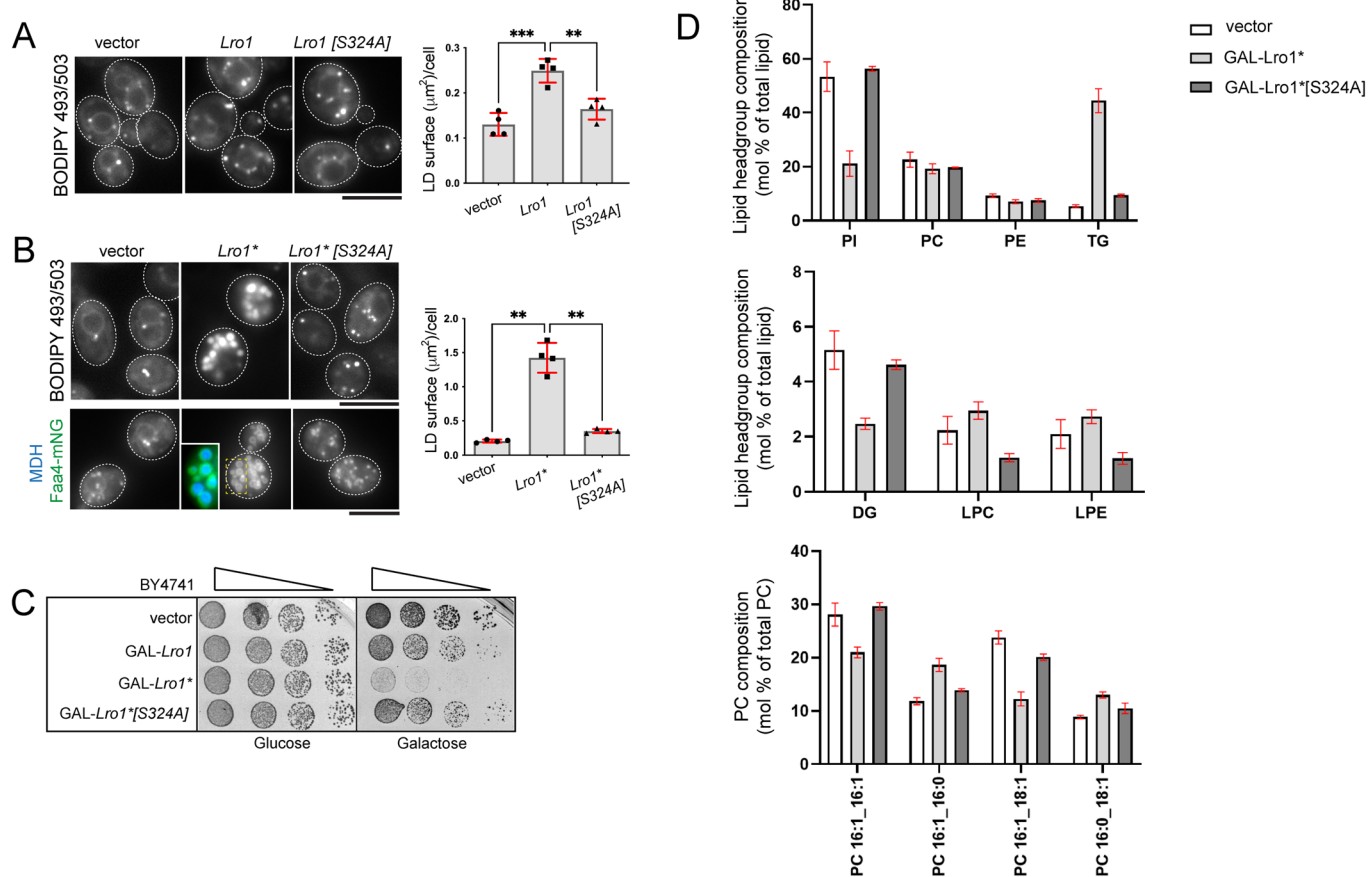

**Figure 3. Elevated PDAT activity disrupts phospholipid and cell homeostasis.**

(A) Wild-type cells carrying an empty vector or the indicated Gal-Lro1 constructs were grown in galactose-containing media for five hours and stained with BODIPY 493/503; data are shown as means ± SD from four experiments with at least 250 cells measured per strain per experiment. Statistical analysis was performed by one-way ANOVA with Šidák correction; **$P < 0.01$, ***$P < 0.001$ (vector vs Lro1 $P = 0.0003$; Lro1 vs Lro1[S324A] $P = 0.0029$). (B) Same as in A but expressing the denoted Gal-Lro1* constructs; lower panels: maximum intensity projections of cells expressing Faa4-mNG; inset shows cells co-stained with monodansylpentane (MDH) to stain LDs; data are shown as means ± SD from four experiments with at least 190 cells measured per strain per experiment. Statistical analysis was performed by Forsythe and Welch ANOVA with Dunnett's T3 correction; **$P < 0.01$ (vector vs Lro1* $P = 0.0037$; Lro1* vs Lro1*[S324A] $P = 0.0055$). (C) Cells carrying the indicated constructs were spotted serially on synthetic plates containing glucose or galactose. Cells were grown at 30 °C and scanned after 2 days. (D) Wild-type cells (BY4741) carrying the indicated constructs were grown in galactose as in (A) and processed for lipidomics analysis as described in Methods. Data are means ± SD from three experiments. Scale bars in all micrographs, 5 μm. Source data are available online for this figure.

## Molecular dynamics simulations suggest that the luminal linker controls lipid accessibility to the catalytic domain of Lro1*

Next, we investigated the mechanism by which the luminal linker of Lro1 inhibits its PDAT activity. The entire luminal linker could sterically hinder lipid access to the catalytic site of Lro1; alternatively, increased PDAT activity in Lro1* could result from the removal of a specific inhibitory sequence within the Ser104 to Asp152 fragment. To distinguish between these models, we replaced Ser104 to Asp152 with a random Gly/Ser stretch of the same length (Fig. 4A). This "GS string" Lro1 mutant was functional as it supported LD formation at the stationary phase and rescued the oleate-induced lipotoxic death of 4Δ (Fig. 4B,C). However, it failed to phenocopy Lro1*, i.e., to induce the formation of large LDs in the exponential phase of growth or rescue the lipotoxic phenotype of 4Δ at higher concentrations of oleate (Fig. 4B,C).

Therefore, the elevated PDAT activity in Lro1* is not due to the lack of a specific amino acid sequence within the removed sequence.

To gain further insight into how the removal of the luminal helices affects Lro1 activity, we performed coarse-grained molecular dynamics (CGMD) and all-atom molecular dynamics simulations in a model phospholipid membrane (see "Methods"). The data show that the core soluble domain of Lro1 and Lro1* is similarly dynamic in both structures. In wild-type Lro1, the luminal linker segment retains overall its secondary structure during the simulations and is also relatively stable (Appendix Fig. S7A,B). The predicted helices keep the lipids planar within the membrane and there is no phospholipid extraction towards the catalytic site during the simulations (5 repeats of 1 μs); on the other hand, in Lro1* there is clear extraction of phospholipids as indicated by the deformation of the model membrane that faces the enzyme, observed in all five repeats of Lro1* and illustrated for one repeat in

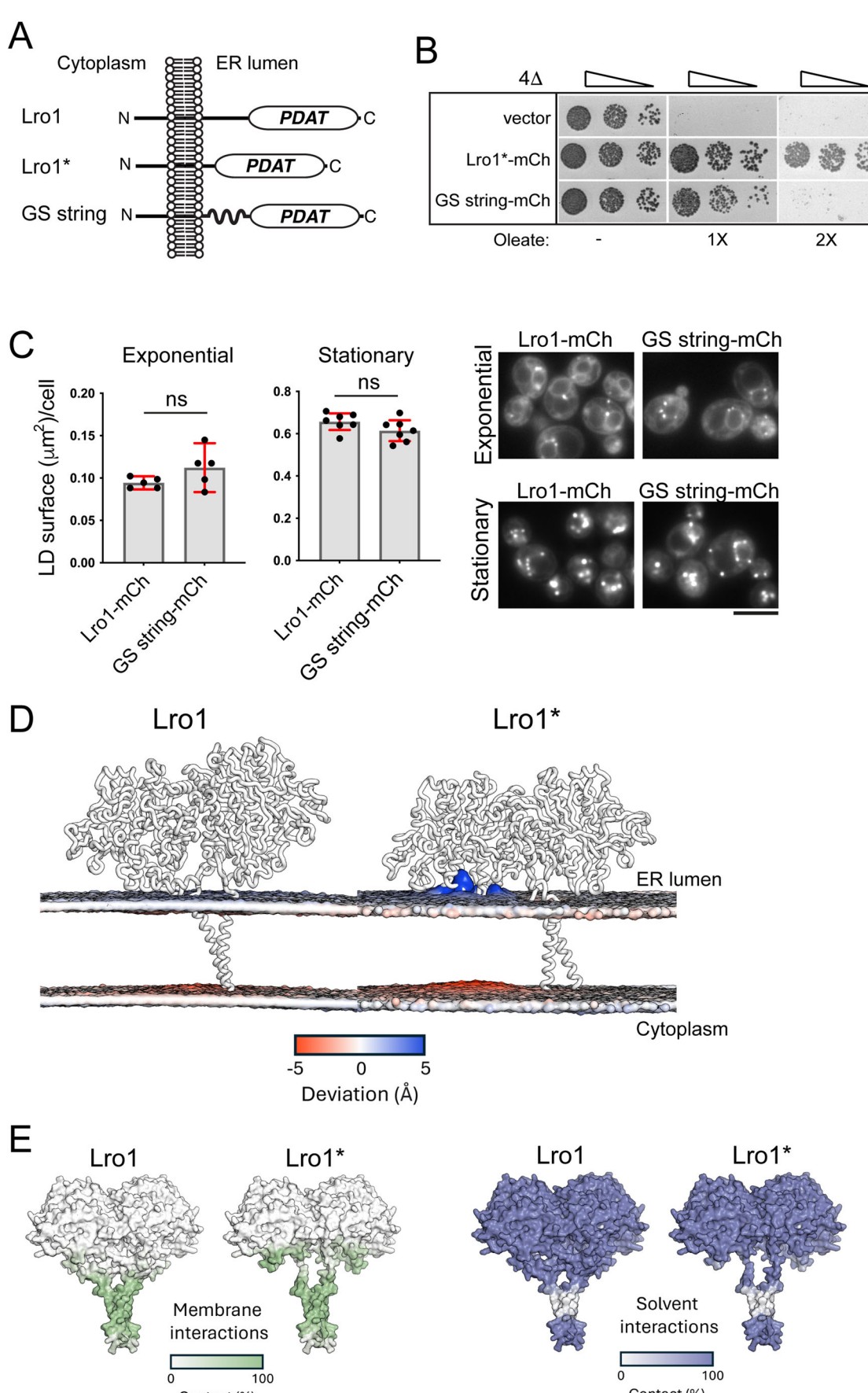

**Figure 4.  Molecular dynamics simulations of Lro1*.**

(A) Schematic and topology of Lro1, Lro1* and Lro1 GS string mutant. (B) 4Δ cells carrying an empty vector or the indicated constructs were grown in selective media and spotted serially on YEPD plates in the absence or presence of two different concentrations of oleate. Cells were grown at 30 °C for 2 days. (C) 4Δ cells, expressing Lro1 or Lro1 GS string, were stained with BODIPY 493/503 in the exponential or stationary phases and the LD size was quantified. Data shown as mean ± SD of five (exponential) or seven (stationary) experiments with at least 200 cells measured per strain per experiment and per growth phase. Welch's t test was performed for cells in exponential growth while unpaired t-test was performed for cells in stationary phase; ns, not significant; right panels: representative cells labeled by BODIPY 493/503 are shown; scale bar, 5 μm. (D) Coarse-grained molecular dynamics simulations in model phospholipid membrane. Changes in membrane distortions are observed when comparing Lro1 and Lro1*, which are shown as a gray surface representation. In Lro1, the lipid bilayer remains planar while in Lro1* the lipid headgroups access the active site. (E) Membrane and solvent interactions of Lro1 and Lro1* shown as color representations. Lro1* shows more contacts with the membrane. Source data are available online for this figure.

Fig. 4D. Three repeats show consistent membrane deviations, while in two repeats this is only transient. We further assessed the proximity of the abstraction to the active site of Lro1, by measuring the percentage of time a lipid is within 6 Å from the active site residue His618. In the three repeats that showed the greatest deformation, a lipid was in contact with His618 for 40.26%, 37.86%, and 27.37% of the simulation time. In the two other repeats this contact was far briefer at 0.50% and 6.09% of the simulated duration. In none of the Lro1 repeats was a lipid within 6 Å of His618.

Chai-1 predictions confirm the binding of both the substrate (DG) and product (TG) to the proposed catalytic triad in wild-type Lro1 (Appendix Fig. S7C,D) (bioRxiv: Chai Discovery and Boitreaud, 2024). Furthermore, the simulations show that the catalytic domain of Lro1* forms more extensive contacts with the membrane in the absence of the helical segments (Fig. 4E). Taken together, these data support the model in which the Ser104–Asp152 region controls the accessibility of the phospholipid substrate to the catalytic site of Lro1; and that in its absence, Lro1 becomes constitutively active.

## Genetic requirements for survival in cells with elevated PDAT activity

The rapid proliferation of yeast cells requires phospholipid synthesis in order to support membrane growth. Lro1*-expressing cells, however, do not show growth defects suggesting that they compensate the effects of elevated PDAT activity by maintaining membrane phospholipid pools at physiological levels. We reasoned that increasing levels of (a) de novo phospholipid synthesis, (b) turnover of FAs from LDs via lipolysis and/or autophagy, or (c) lysoPL reacylation, may maintain membrane homeostasis in Lro1*-expressing cells (Fig. 5A). Accordingly, the absence of these pathways could be detrimental in cells with elevated PDAT activity. To test this hypothesis, we used the copper-inducible Cup1 promoter to moderately overexpress Lro1* in a select set of mutants (Fig. 5B). Compromising glycerolipid synthesis (gpt2Δ, sct1Δ), macroautophagy (atg1Δ), lipophagy (atg15Δ) or lipolysis (tgl3Δ tgl4Δ tgl5Δ) did not stop the growth of Lro1*-expressing cells (Fig. 5B). Moreover, atg1Δ did not affect LD formation, indicating that autophagy is not required to recycle FAs for the formation of large LDs by elevated PDAT activity (Fig. 5C). Activation of PC synthesis through the Kennedy pathway, induced by the addition of choline, resulted in a modest reduction in LD content in Lro1*-expressing cells; however, these cells still contained significantly more LDs than Lro1-expressing cells (Appendix Fig. S8). Removal of the broad range lysoPL

acyltransferase (LPLAT) Ale1 (Benghezal et al, 2007; Chen et al, 2007; Riekhof et al, 2007; Tamaki et al, 2007), which is involved in post-synthetic phospholipid remodeling, was lethal in cells expressing Lro1* (Fig. 5B,D). Loss of the second major LPLAT, Slc1, had not the same effect (Fig. 5B). This finding suggests that Ale1 is essential to maintain PE and PC levels in the presence of elevated PDAT activity and is the likely cause for the changes in PC molecular species in the Lro1*-expressing cells (Fig. 3D).

## PDAT activity mediates the turnover of the ER membrane

The availability of a hyperactive Lro1 enzyme allowed us to address directly the function of PDAT activity in membrane biogenesis. The conversion of PA to DG, catalyzed by Pah1 and counteracted by Dgk1, is the key step that controls the partitioning of precursors between phospholipids and TG (Fig. 6A) (Henry et al, 2012). Shifting the balance towards PA, either by inactivating Pah1 or by overexpressing Dgk1, results in increased phospholipid synthesis and membrane expansion (Han et al, 2008a; Han et al, 2006; Santos-Rosa et al, 2005). Indeed, mitotic phosphorylation of Pah1 decreases its membrane association, resulting in increased PA levels at the nuclear envelope and nuclear membrane expansion to facilitate the "closed mitosis" of yeast (Choi et al, 2011; Saik et al, 2023; Santos-Rosa et al, 2005). We previously showed that Lro1 concentrates to a INM subdomain that is known to expand in response to phospholipid synthesis; and that during mitosis, Lro1 redistributes from that nuclear subdomain throughout the ER (Barbosa et al, 2019). We therefore postulated that the PDAT activity of Lro1 negatively regulates membrane expansion.

To test this hypothesis, we employed an in vivo assay to induce "ectopic" ER membrane expansion and then determine how Lro1* influences it. We used a strain in which Dgk1 is overexpressed with the inducible Gal1/10 promoter and Lro1* with the inducible Cup1 promoter (Fig. 6B). As expected, following the addition of galactose, these cells display membrane expansion characterized by deformed nuclei and large Sec63-mNG membrane sheets that fill the cytoplasm and can be detected in both cortical and middle sections (Fig. 6C). Transferring the cells to glucose, in the presence of copper to induce Cup1-Lro1*, resulted in significant restoration of wild-type ER membrane morphology (Fig. 6C,D), without affecting protein levels of Dgk1 (Appendix Fig. S9). The restoration of normal membrane morphology required the catalytic activity of Lro1* and, importantly, coincided with the appearance of large LDs in the Lro1*-expressing cells (Fig. 6C–E). To further examine the dynamics of this process, we performed time-lapse imaging. Shortly after the induction of Lro1*, we observed the progressive

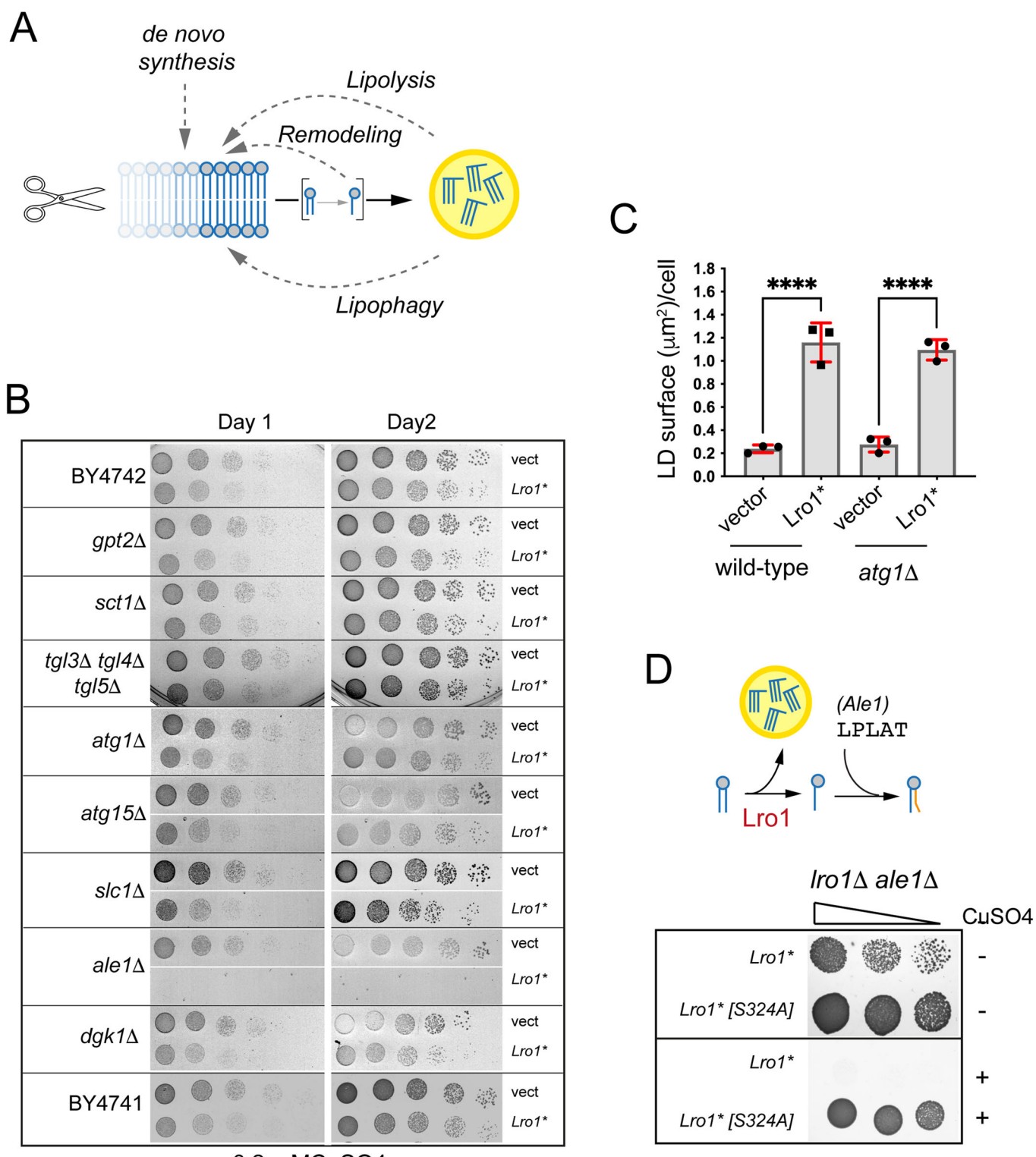

shortening of the proliferated ER membrane sheets followed by the reappearance of normal ER morphology (Fig. 6F); cells with brighter Sec63-mNG-containing ER sheets, likely reflecting the association of multiple stacked ER membranes together, were often partially retracting during the course of the time-lapse experiment

(Appendix Fig. S10). Cumulatively, these results show that PDAT activity can mediate the retraction of proliferated ER membranes.

The major pathway that clears membrane-bound organelles is autophagy. Lro1* was able to restore normal ER morphology in *atg1*Δ cells, which are deficient in receptor-mediated macro-ER

**Figure 5. A targeted screen for genetic interactors of Lro1\*.**

(A) Schematic of pathways that could compensate elevated PDAT activity in yeast. (B) Wild-type (BY4741 or BY4742) or the denoted mutants, carrying an empty vector or a high-copy plasmid expressing Lro1\* under the control of the Cup1 promoter were spotted on synthetic plates containing copper sulphate and scanned after one or 2 days. (C) Quantification of the BODIPY 493/503 labeling of the denoted strains. Data are means ± SD from three experiments with at least 160 cells measured per strain per experiment. Statistical analysis was performed by one-way ANOVA with Šidák correction; ****$P < 0.0001$. (D) Upper panel: schematic of the activities of Lro1 and Ale1; lower panel: the *lro1Δ ale1Δ* mutant carrying low-copy plasmids expressing the denoted Lro1\* alleles under the control of the Cup1 promoter, was spotted on plates with or without copper sulphate. Source data are available online for this figure.

autophagy (Fig. 6G). Moreover, the fraction of cells with Sec63-mNG-containing membrane whorls within the vacuole was not different between Lro1\* and control cells during the time course of Lro1\* expression (Fig. 6H). Taken together, these observations support the conclusion that the PDAT-mediated turnover of the ER membrane is independent of Atg1-meciated macro-ER autophagy.

## PDAT activity is induced by Pah1-generated DG and clears the ER membrane

PDAT activity requires DG as a fatty acyl acceptor for the synthesis of TG. However, DG is also utilized by Dgk1 for phospholipid production and the expansion of the ER (Fig. 6A). This raises the question of what is the source supplying the DG necessary for the retraction of the expanded ER membrane by Lro1\*. Notably, following the induction of Dgk1 in wild-type cells, we observed that Pah1 was recruited on the ER membrane, suggesting that it is providing DG for the PDAT reaction (Fig. 7A). If this is the case, Lro1\* would not be able to revert the ER membrane expansion of cells lacking Pah1. Elevated PA levels in *pah1Δ* cells result in nuclear/ER membrane expansion, which is similar to that seen in Gal-Dgk1 cells (Han et al, 2008a; Han et al, 2008b). Indeed, and contrary to Gal-Dgk1-expressing cells, the proliferated ER membrane in *pah1Δ* cells was not retracted by Lro1\* (Fig. 7B). Therefore, Pah1 is required for Lro1\* to turnover excess ER membrane.

Next, we asked whether the production of DG by Pah1 controls PDAT activity. To elevate DG levels, we expressed the dephosphorylated and constitutively active version of Pah1, Pah1-7A. As previously shown, expression of Pah1-7A inhibits cell growth (O'Hara et al, 2006); and this inhibition is rescued by the addition of inositol and choline in the medium, which alleviate the inhibitory effects of low PA and high DG levels, respectively (Choi et al, 2011) (Fig. 7C). However, growth of Lro1\*-expressing cells was still inhibited in the presence of inositol and choline, in a manner that depended on its catalytic PDAT activity (Fig. 7C). We conclude that when DG levels increase, PDAT activity becomes detrimental to cells.

We next sought to determine how DG and PDAT activity influence ER membrane organization. Cells with either elevated PDAT activity (Lro1\*, expressed under the control of Lro1 promoter), elevated DG levels (Pah1-7A, expressed under the control of the Gal1/10 promoter), or with both combined, were visualized by super-resolution microscopy. To evaluate ER membrane morphology quantitatively, we developed a semi-automated tool that measures cortical ER membrane area from mid-sections (see "Methods"). Our analysis showed that expression of Lro1\* alone does not significantly alter ER membrane area of wild-type cells (Fig. 7D). However, when DG levels increased in

Lro1\* cells, the ER membrane area was significantly reduced (Fig. 7D,E). Inspection of optical z sections obtained by super-resolution microscopy (Appendix Fig. S11) and three-dimensional reconstruction of these cells (Fig. 7F) revealed extended cortical areas entirely devoid of ER membrane. Despite this decrease in ER membrane area, LD content in Pah1-7A/Lro1\*-expressing cells showed a modest but significant rise when compared to Lro1\*-expressing cells (Fig. 7G). The plasma membrane, which in yeast is closely apposed to the cortical ER, was not disrupted in Pah1-7A/Lro1\*-expressing cells (Fig. 7H). Taken together, these observations support the notion that the availability of DG controls PDAT activity within the ER membrane.

Based on the observations that elevated PDAT activity decreases (a) DG levels and (b) the nucleolar enrichment of Lro1\*, we asked whether DG produced by Pah1 controls the distribution of Lro1. Consistent with this hypothesis, in *pah1Δ* cells, wild-type Lro1 was no longer enriched in the nucleolar INM subdomain during the G1/S phase (Fig. 8A). Furthermore, inducing briefly the expression of Pah1-7A (3 h) increased the pool of nucleolar Lro1\* (Fig. 8B,C); we noticed that upon longer induction of Pah1-7A, this effect was lost, possibly due to the concurrent damage to the ER membrane caused by Lro1\* disrupting its movement to the INM. Collectively, these results indicate that DG levels, controlled by Pah1, regulate the Lro1 PDAT activity and its partitioning between the nuclear and cytoplasmic ER membrane network.

## Discussion

PDATs are atypical DG acyltransferases in the sense that they use organelle membranes as a source of FAs to generate TG. Unlike DGATs, which have well studied roles in FA handling, the biological functions of PDATs have remained elusive.

In this work, we exploited an engineered version of Lro1, Lro1\*, to investigate the function of PDAT activity in vivo. Lro1 is predicted to contain two short helices which lie parallel and in close proximity to the luminal phospholipid leaflet of the ER membrane. Crucially, the helices are also positioned over the active site of Lro1 (Fig. 1C). This configuration is similar to what has been described as a "lid" domain in a number of soluble prokaryotic and eukaryotic lipases (Khan et al, 2017). Lids can vary from loops, one or two helices and shift between a closed and an open conformation to block or allow access to the substrate binding site, respectively. Our MD simulations show that the removal of this inhibitory sequence results in the catalytic triad of the Lro1\* dimer gaining access to the ER membrane and consequently to its phospholipid and DG substrates, thereby accelerating the PDAT reaction. Consistently, the effects of Lro1\* on LDs and the ER membrane required its catalytic activity. Hence, we propose that

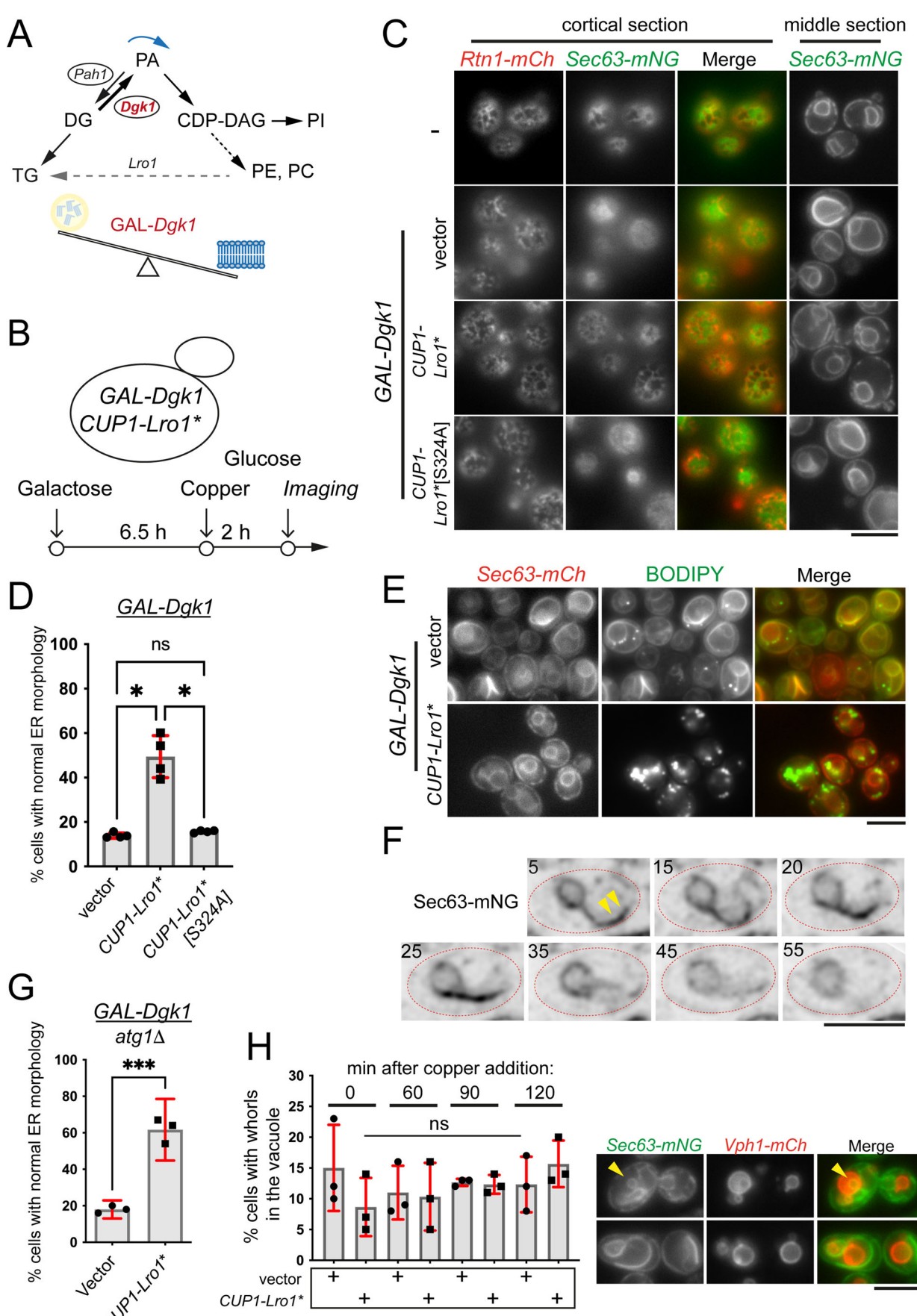

**Figure 6. PDAT activity counteracts ER membrane expansion.**

(A) Role of Pah1 and Dgk1 activity in phospholipid synthesis. (B) Schematic of the experimental setup used; yeast cells expressing (a) an ER reporter (Sec63-mCh, Sec63-mNG or Sec63-mNG together with Rtn1-mCh), (b) a Gal1/10-inducible *Dgk1*, or an empty vector, and (c) a Cup1-inducible *Lro1\**, or *Lro1\**[S324A] or an empty vector, were grown in galactose-containing media for 6.5 h to induce ER membrane expansion and then transferred to medium containing glucose and 0.2 mM copper sulphate to induce Lro1\*. (C) Representative ER morphology at the end point of the assay as described in (B) ("Imaging"); both cortical and middle sections of wild-type (top panels) or Gal-Dgk1-overexpressing cells (lower panels) carrying the indicated plasmids are shown. (D) Quantification of Sec63-mCh-expressing cells with normal ER morphology at the end point of the assay; data are means from four experiments ± SD with at least 130 cells measured per strain per experiment. Statistical analysis was performed by Forsythe and Welch ANOVA with Dunnett's T3 correction; * for $P < 0.05$ (vector vs Cup1-Lro1\* $P = 0.012$; Cup1-Lro1\* vs Cup1-Lro1\*[S324A] $P = 0.0136$). (E) Cells at the end point of the assay were stained with BODIPY to label LDs. (F) Time-lapse imaging of cells grown as detailed in (B), to express sequentially Gal-Dgk1 and Cup1-Lro1\*. The sequence starts after cells were growing for 50 min in glucose/copper sulphate-containing media, corresponding to time point 0 of the time-lapse. Individual frames and the time points (in min) when they were captured are shown; arrowheads point to the expanded ER; the dotted lines indicate the cell contours. (G) *atg1Δ* cells expressing Sec63-mCh were processed as described in B and the percentage of cells with normal ER morphology at the end point of the assay was calculated. Data are means ± SD from three experiments with at least 90 cells were measured per strain per experiment. Statistical analysis was performed by an unpaired *t* test; ***$P < 0.001$ (vector vs Cup1-Lro1\* $P = 0.0004$). (H) Cells co-expressing Sec63-nNG and Vph1-mCh were processed as shown in B and the percentage of Sec63-mNG-containing whorls inside the vacuole during the time-course was quantified. Data are means ± SD from three experiments with at least 70 cells measured per strain per experiment. Statistical analysis was performed by one-way ANOVA; ns, not significant; right panel: representative cells with (top panels) or without (lower panels) vacuolar ER whorls; the arrowhead points to an example of a vacuolar ER whorl. Scale bars in all micrographs, 5 µm. Source data are available online for this figure.

Lro1\* recapitulates the active state of PDATs in vivo and is a powerful tool to study their biology. According to our model, a conformational rearrangement will be required for the wild-type Lro1 to displace the two helices and allow the catalytic reaction to proceed. Helix positioning could be regulated by lipid composition, such as the levels of DG, the oligomerization state of Lro1, or its interaction with other protein components. Unlike soluble lipid-modifying enzymes, which are regulated by their reversible association with membranes (Cornell and Taneva, 2006), Lro1 is membrane-bound and therefore tight control of its activity could be important to avoid the deleterious effects of deregulated phospholipid deacylation. However, we cannot rule out that the removal of the helical region induces the activity of Lro1 through alternative mechanisms. Elucidating the underlying processes will eventually require determining the structure of Lro1 and its mode of interaction with the phospholipid bilayer.

We show here that a key determinant of PDAT activity is DG generated by Pah1. Firstly, extended ER membrane sheets, formed by excess phospholipids, can be retracted by Lro1\* only if an active, membrane-associated, Pah1 is present; secondly, once DG availability is no longer limiting, due to the constitutive activation of Pah1, elevated PDAT activity mediates the uncontrolled breakdown of endogenous cortical ER. These findings support a model where Pah1 and PDAT activities are coordinated to inhibit ER membrane biogenesis. We speculate that this mechanism, coupling DG production to PDAT activation, applies locally at the nucleolar-associated membrane. This subdomain is targeted by Lro1 and is known to expand in response to excess phospholipid synthesis (Barbosa et al, 2019; Witkin et al, 2012). Mitotic phosphorylation of Pah1 downregulates its membrane association, increasing PA levels at the nuclear membrane and phospholipid synthesis for nuclear membrane expansion. Concomitantly, Lro1 is displaced from this subdomain and redistributes to the ER. As cells enter the new cycle, dephosphorylation of Pah1 restores DG production, coinciding with the relocation of Lro1 at the nuclear membrane subdomain. Consistent with this hypothesis, we find that Pah1-derived DG promotes the INM targeting of Lro1\* while removal of Pah1 inhibits Lro1 targeting.

The collaboration between Pah1 and Lro1\* to retract the expanded ER membranes raises an interesting topological problem: Pah1 is cytosolic and generates DG on the cytoplasmic leaflet of the membrane (Han et al, 2006) while the catalytic domain of Lro1 faces the ER lumen (Choudhary et al, 2011). Thus, the rate of phospholipid deacylation by Lro1\*, which is likely to be high, must be coupled with the provision of DG at the luminal leaflet where Lro1 resides. The underlying mechanisms are unknown but we propose that they involve the transbilayer (flip-flop) movement of DG which is known to be very rapid and can occur independently of proteins (Contreras et al, 2010).

In addition to Pah1 activity, the second factor regulating Lro1\* is the LPLAT Ale1. The levels of the major PC—and to a lesser extent PE—species are altered in cells overexpressing Lro1\* and Ale1 becomes essential for survival, suggesting that Ale1 re-acylates the lysoPC and lysoPE products of Lro1\*. We have previously shown that the distribution of Ale1 at the nuclear and ER membrane does not change, irrespective of the enrichment of Lro1 at the nucleolar membrane (Barbosa et al, 2019). Increasing the local levels of Lro1 relative to Ale1, in the presence of DG, may therefore favor the accumulation of lysoPLs at the nucleolar membrane subdomain. When Lro1 redistributes to the ER, its combined activity with Ale1 could promote the remodeling of PC. Hence, the availability of DG and LPLAT activity at the different ER membrane subdomains could determine the outcome of PDAT activity.

Unlike Lro1, which dynamically moves between the ER and the INM (Barbosa et al, 2019), Lro1\* localizes at steady state at the ER irrespective of growth or cell cycle stage. The molecular basis of this redistribution remains unknown: removal of the luminal linker segment may prevent the movement of Lro1 through the nuclear pore membrane which is required for its import to the INM; alternatively, the decrease in DG levels, due to the derepressed activity of Lro1\*, could compromise its retention at the INM.

Our data show a major decrease of PI levels following the overexpression of Lro1\* (Fig. 3D) which is detected across all PI species, raising the possibility that PI is a substrate of Lro1. Currently, however, there is no biochemical evidence that Lro1 can directly de-acylate PI. Alternatively, the effects on PI may be indirect: the partitioning of CDP-DG for the synthesis of PI and the PS-generated PE/PC, which is governed by the enzymes Pis1 and Cho1 respectively, may be altered in Lro1\*-overexpressing cells in order to maintain PE and PC levels. Future work will address these possibilities.

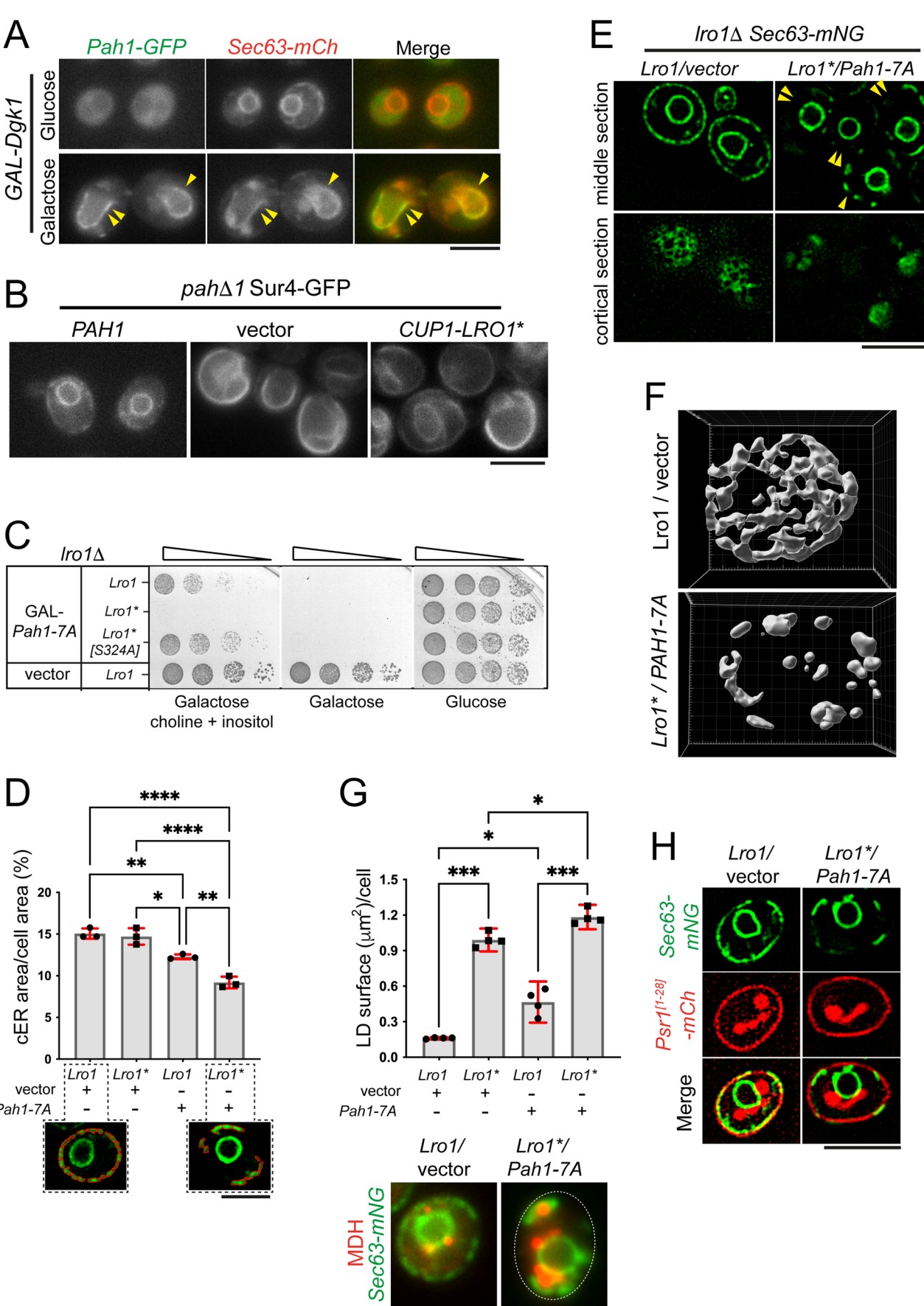

**Figure 7. DG levels control PDAT activity and promote ER membrane degradation.**

(A) Cells co-expressing Pah1-GFP and Sec63-mCh and carrying a Gal-Dgk1 plasmid were grown in glucose or galactose-containing media for 6.5 h; arrowheads point to Pah1-GFP recruited onto the expanded ER membrane. (B) *pah1Δ* cells expressing Sur4-GFP to visualize ER morphology and the indicated plasmids, were grown for 8 h in the presence of 0.2 mM copper to induce Lro1*. (C) *lro1Δ* cells carrying the indicated plasmids were spotted serially on glucose or galactose-containing plates with or without inositol and choline and grown for 3 days at 30 °C. (D) *lro1Δ* cells expressing Sec63-mNG and complemented by Lro1 or Lro1* were transformed with either an empty vector of Gal-Pah1-7A; cells were grown in galactose-containing media for 5 h and imaged by super-resolution microscopy. Quantification of cortical ER area was done as described in "Methods". Data are means ± SD from three experiments with at least 38 cells measured per strain per experiment. Statistical analysis was performed by one-way ANOVA with Šidák correction; *$P < 0.05$, **$P < 0.01$, ****$P < 0.0001$ (vector/Lro1 vs Pah1-7A/Lro1 $P = 0.0032$; vector/Lro1 vs Pah1-7A/Lro1* $P < 0.0001$; vector/Lro1* vs Pah1-7A/Lro1 $P = 0.0127$; vector/Lro1* vs Pah1-7A/Lro1* $P < 0.0001$; Pah1-7A/Lro1 vs Pah1-7A/Lro1* $P = 0.0018$); lower panels: representative examples of cells with segmented cortical ER for area quantification. (E) Representative middle or cortical confocal sections depicting ER morphology of *lro1Δ* cells quantified in (D); arrowheads point to cortical areas lacking ER membrane. (F) Cortical ER membrane morphology of cells expressing Sec63-mNG reconstructed from confocal sections obtained as described in Methods. (G) *lro1Δ* cells complemented by Lro1 or Lro1* and carrying an empty vector of Gal-Pah1-7A were grown as in (D), labeled with BODIPY and LDs were quantified. Data are means ± SD from four experiments with at least 140 cells measured per strain per experiment. Statistical analysis was performed by Forsythe and Welch ANOVA with Dunnett's T3 correction; *$P < 0.05$, *** for $P < 0.001$ (vector/Lro1 vs vector/Lro1* $P = 0.0004$; vector/Lro1 vs Pah1-7A/Lro1 $p = 0.0423$; vector/Lro1* vs Pah1-7A/Lro1* $P = 0.0234$; Pah1-7A/Lro1 vs Pah1-7A/Lro1* $P = 0.0005$); lower panels: *lro1Δ* cells expressing Sec63-mNG and the denoted plasmids were grown as in (D) and labeled with MDH; the dotted line indicates the cell contour. (H) Wild-type cells expressing Sec63-mNG and the 28 amino-terminal residues of Psr1 fused to mCherry to visualize the plasma membrane (Siniossoglou et al, 2000) and the denoted plasmids, were grown and imaged as in (D); note that Psr1[1-28]-mCh accumulates also in the vacuole. Scale bars in (A, B, E, H), 5 µm; in (D, G), 3 µm. Source data are available online for this figure.

How does PDAT activity retract the expanded ER membranes? The main degradative pathway which clears membrane-bound organelles is autophagy. PDAT-mediated phospholipid remodeling may facilitate the uptake of excess ER membrane in the vacuole; alternatively, highly active PDAT could damage the ER membrane and consequently trigger its degradation by ER-phagy (Reggiori and Molinari, 2022; Schuck, 2020). We found no evidence of increased Sec63-mNG-containing membranes in vacuoles of cells expressing Lro1*; moreover, retraction of proliferated ER membrane took place in cells lacking Atg1. Therefore, we favor a model where Lro1* drives the breakdown of the ER membrane independently of the core macro-autophagic machinery. However, we cannot exclude the contribution of Lro1* to micro-autophagy which can degrade the ER in an Atg1-independent pathway (Schäfer et al, 2020). Alternatively, Lro1 could de-acylate both fatty acyl chains from its substrate; a second possibility is that the lysoPC/PE product of Lro1 is targeted by additional ER-localized phospholipases. Degradation of membrane-bound organelles has been recently reported by members of the mammalian phospholipase A/acyltransferases (PLAATs) family during lens development (Morishita et al, 2021). Engineered PLAATs can also degrade mitochondria and peroxisomes (Watanabe et al, 2022). In these cases, however, the fate of the released fatty acyl chains remains unknown. In the case of PDATs, the ability to store the deacylated FAs into LDs prevents the localized buildup of excess free FAs. Hence, this would allow cells to use PDAT activity to turnover organelle membranes avoiding lipotoxic stress.

By mobilizing FAs from membranes to LDs in an acyl-CoA-independent mechanism, PDATs could remodel organelle size and shape during the cell cycle or development, when cells undergo significant changes. Therefore, the use of membrane lipids as FA donors for TG synthesis may have evolved primarily as a tool to remodel organelles rather than to store metabolic energy in LDs during stress. The fact that green algal PDATs use chloroplast lipids as their substrates indicates the diversity of organelle-specific functions for PDATs (Yoon et al, 2012). Although humans lack clear PDAT orthologues, the recent identification of a pathway that generates TG at the expense of membrane phospholipids indicates the operation of a similar process. It will be interesting to define

how these acyl-CoA-independent TG biosynthetic pathways regulate organelle biogenesis and cellular physiology.

## Methods

### Reagents and tools table

| Reagent/resource | Reference or source | Identifier or catalog number |
|---|---|---|
| **Experimental models** | | |
| *Saccharomyces cerevisiae* | Open Biosystems | BY4741 and derivatives—Appendix Table S3 |
| **Recombinant DNA** | | |
| DNA plasmids | Appendix Table S1 | Appendix Table S1 |
| **Antibodies** | | |
| HRP Goat Anti-Rabbit IgG | BD Biosciences | 554021 |
| Anti-Dgk1 | Han et al, 2008b | Han et al, 2008b |
| **Oligonucleotides and other sequence-based reagents** | | |
| Oligonucleotides | Appendix Table S2 | Appendix Table S2 |
| **Chemicals, enzymes, and other reagents** | | |
| BODIPY 493/503 | Invitrogen | D3922 |
| Oleic Acid | Sigma Aldrich | 05508 |
| Monodansylpentane (MDH) | Abcepta | SM1000a |
| Water for UHPLC suitable for mass spectrometry (MS) | Sigma Aldrich | 900682-1L |
| Acetonitrile, MS SupraSolv | Sigma Aldrich | 1006651000 |
| Methanol, HPLC grade | Sigma Aldrich | 34860-1L-R |
| Chloroform | Sigma Aldrich | C2432-1L |
| Isopropanol Optima LC/MS Grade | Fisher Scientific | 10091304 |
| Ammonium formate | Sigma Aldrich | 516961-100G |
| SPLASH LpidoMIX Mass Spec Standard | Avanti Polar Lipids | 330707 |
| **Software** | | |
| Image J | https://imagej.net/ij/ | ImageJ 2.1.0/1.53c |

| Reagent/resource | Reference or source | Identifier or catalog number |
|---|---|---|
| Imaris | https://imaris.oxinst.com/ | Imaris 10.1.0 |
| Prism | https://www.graphpad.com/ | Prism 10.2.3 |
| Adobe Photoshop | https://www.adobe.com/ | Photoshop 2024 |
| Other | | |

## Plasmids

Plasmids are listed in Appendix Table S1. Oligonucleotide DNA sequences are listed in Appendix Table S2. Lro1 constructs were cloned into the YCplac/YEplac vectors unless specified otherwise (Gietz and Sugino, 1988). All Lro1/Lro1* constructs contained a BamHI restriction site immediately prior to the stop codon, which was used to insert fragments encoding various epitope tags. Lro1* constructs were generated by replacing the sequence coding for 49 amino acid residues, from Ser104 to Asp152, with a BamHI restriction site. The Lro1 GS string construct was generated by introducing 45 residues from the glycine-serine-rich domain of the bacteriophage M13 attachment protein G3P (accession number P69168) into Lro1*, flanked by two BamHI sites. The sequence is: GS-VNAGGGSGGGSGGGSEGGGSEGGGSEGGGSEGGGSGGG SGSGDFD-GS.

## Yeast strains, media, and growth conditions

Strains are listed in Appendix Table S3. Unless indicated otherwise, all strains were derived from the BY4741 or BY4742 backgrounds. Gene deletions and epitope tagging by chromosomal integration were generated by a one-step PCR-based method (Janke et al, 2004;

Longtine et al, 1998) and confirmed by PCR. Tagging of Sec63 and Rtn1 with mNeonGreen (mNG) and mCherry (mCh), respectively, was done by using as a template the genomic DNA of a yeast strain expressing these tagged genes (Papagiannidis et al, 2021) and transforming the PCRs into BY4741. Yeast cells were transformed using the lithium acetate method. Cells were cultured in synthetic medium (SC) comprising 2% glucose, 0.17% yeast nitrogen base (Difco, BD, Franklin Lakes, NJ), 0.5% ammonium sulfate, and amino acid drop-out. The drop-out contained 55 mg/l uracil, 40 mg/l tryptophan, 55 mg/l adenine, 10 mg/l methionine, 10 mg/l histidine, 60 mg/l leucine, 60 mg/l isoleucine, 60 mg/l phenylalanine, 55 mg/l tyrosine, 50 mg/l threonine, 40 mg/l lysine, and 20 mg/l arginine, lacking the specific amino acids required for plasmid selection. Cells were grown overnight to the exponential phase ($OD_{600nm}$ 0.4–0.6) or to the post diauxic shift (PDS) phase ($OD_{600nm}$ 4–6) from fresh precultures.

For genes expressed from the Cup1 promoter, 0.2 mM of CuSO4 was added to the media for the specified amount of time. For genes expressed from the Gal1/10 promoter, precultures were grown over the day in media containing 2% glucose; cells were then washed twice with water and transferred to media containing 2% raffinose for overnight growth; next day, cells were transferred to media containing 2% galactose for the specified amount of time.

For dot spot growth assays, log phase cells from liquid cultures were diluted to $OD_{600nm}$ 0.2. Serial dilutions (5x) were spotted on the appropriate plates and incubated at 30 °C for 1 to 3 days. Inositol (75 μM) and choline (1 mM) were added to the medium where specified. Oleate was added to YEPD plates at 1 mM or 2 mM with 1% tergitol, as specified (1× or 2×, respectively).

## Epifluorescence microscopy

Strains grown at 30 °C to the indicated growth phases were pelleted and promptly imaged live at room temperature with a Zeiss AxioImage Z2 epifluorescence upright microscope equipped with a 100× Plan-Apochromatic 1.4 numerical aperture (NA) objective

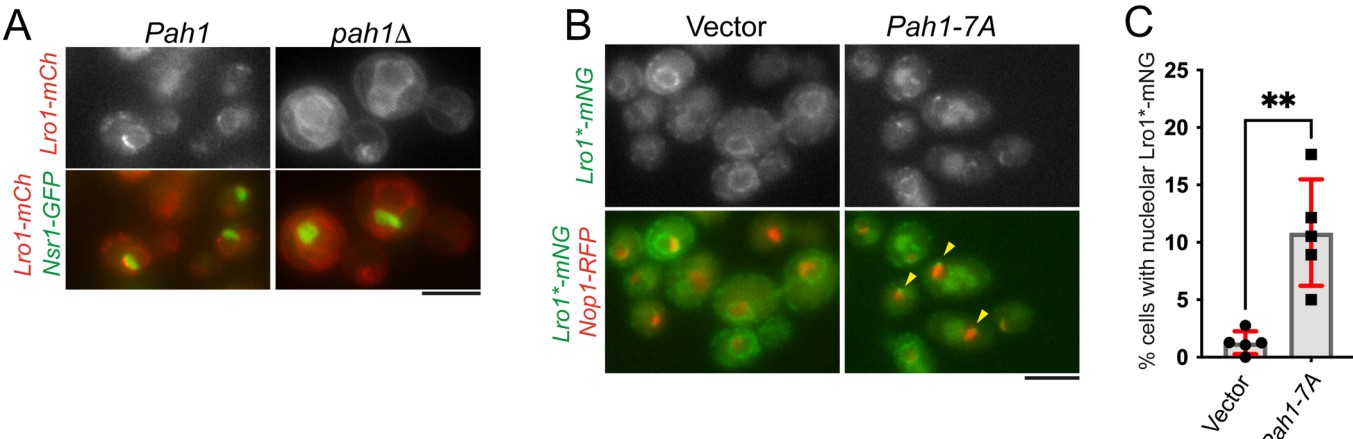

**Figure 8.  The distribution of Lro1 within the ER membrane is controlled by DG levels.**

(A) Distribution of Lro1-mCh in wild-type (*Pah1*) or *pah1Δ* cells expressing a chromosomally integrated Nsr1-GFP to visualize the nucleolus. (B) Distribution of Lro1*-mNG in *lro1Δ* cells expressing Nop1-RFP to visualize the nucleolus and the denoted plasmids; cells were grown in galactose-containing media for 3 h; arrowheads point to Lro1*-mNG in close proximity to the nucleolus. (C) Percentage of cells that show the association of Lro1*-mNG with the nucleolus following the expression of Pah1-7A; data are means of five experiments ± SDs with at least 140 cells measured per strain per experiment. Statistical analysis was performed by Welch's *t* test; **$P < 0.01$ (vector vs Pah1-7A $P = 0.0087$). Scale bars in all micrographs, 5 μm. Source data are available online for this figure.

lens (Carl Zeiss, Jena, Germany). Image acquisition was conducted using a high-sensitivity fluorescence imaging large chip sCMOS mono camera (ORCA Flash 4 version 2; Hamamatsu, Hamamatsu, Japan). Zeiss ZEN blue software was employed to save raw image files, which were then exported to Photoshop (Adobe, San Jose, CA).

Lipid droplets were stained with either 1.25 µg/ml BODIPY 493/503 or 10 µM monodansylpentane (MDH) for 10 min at room temperature.

For ER membrane retraction experiments (Fig. 6), cells carrying Gal-Dgk1 and Cup1-Lro1* or control plasmids were first transferred to medium containing galactose and grown for 6.5 h. Cells were then transferred to medium containing glucose and 0.2 mM CuSO₄ and grown for another for 2 h. For time-lapse imaging of the same strains, cells were grown for 8 h in galactose media, transferred for 50 min in glucose/CuSO4, mounted in agarose pads containing glucose/CuSO₄ and imaged for another hour.

Nucleolar localization of Lro1*-mNG (Fig. 8) was estimated based on its co-localization with Nop1-RFP and the characteristic crescent shape taken by Lro1 when localized to the nucleolus. To account for the 3D shape of the nucleus, we checked different depths across the through-focus series.

## Dual-iterative lattice structured illumination microscopy (super-res SIM2)

Cells were prepared as above. Images were acquired using an Elyra 7 microscope (Carl Zeiss Ltd) using a 64x/1.4NA Plan-Apochromat lens and 488 nm or 561 nm diode lasers and Z-stack images were captured on a pair of PCOedge4.2 sCMOS cameras (Excelitas Technologies). The raw data was processed for SIM2 with Zen Black 3.0SR software. We captured approximately 20 Z-stacks—the exact number depending on the detection of the top and bottom of cells—with a thickness of 0.316 µm and leap settings. Raw images were processed using Zen Black software with the following settings: lattice SIM, leap mode Sim² Standard live, 16 iterations, regulation weight of 0.065, and histograms scaled to the raw image.

## ImageJ analysis

Micrographs obtained by fluorescence microscopy were analyzed and quantified using custom-made macros in Fiji (Image J, version 2.1.0/1.53c) (Schindelin et al, 2012). These are described below.

### Quantification of LD size
To overcome problems caused by the tendency of BODIPY to stain membranes when LDs are small in size, two different macros were written and used to detect and measure lipid droplet size. Quality and accuracy of data obtained macros were exanimated post hoc.

**Big droplet macro:** The micrograph was first processed using the subtract background with rolling ball method of 50 pixels and then turned into 8-bit. This step was followed by 10 pixels rolling ball background subtraction. To further separate LDs from the background, the unsharp mask function was used with a radius of 3 pixels and mask weight of 0.9. Then the threshold for pixel intensity was set between 100 and 255 and the processed image was turned into mask. The watershed function was then applied to separate LD conglomerates. Subsequently, the analyze particles function was run with the range for pixel size set up between 10 and 1000 pixels.

**Small/no droplet macro**: The micrograph was first processed using the subtract background with rolling ball method of 50 pixels and then turned into 8-bit. This step was followed by 5 pixel rolling ball background subtraction. To further separate LDs from the background, the unsharp mask function was used with a radius of 3 pixels and mask weight of 0.9. Then the threshold for pixel intensity was set to 255 and the processed image was turned into mask. Subsequently, the analyze particles function was run with the range for pixel size set up between 10 and 1000 pixels and circularity from 0.6 to 1.

In both macros, the outline of the detected LDs was overlayed with the original image to confirm the accuracy of the detections. Area measurements were then calculated and expressed as LD surface per cell.

### Semi-automated measurement of cortical ER membrane area
Images were acquired using Dual-Iterative Lattice Structured Illumination Microscopy (Super-Res SIM2). Maximum Intensity Projections (MIP) were generated from three slices of a Z-stack, focusing on nuclei. Twenty cells were randomly selected from two fields each and cropped. Subsequently, a macro was applied to measure cortical ER area as follows: the macro was designed to create two masks. Initially, a median filter with a radius of 1 pixel was applied, followed by background subtraction using a rolling ball of 10 pixel. An unsharp mask with a radius of 5 pixels and a mask weight of 0.6 was then applied. Signal detection used autothresholding with RenyiEntropy. Detected signals formed the first mask, followed by fill holes and watershed operations for accurate segmentation. The second mask focused on capturing thinner filaments. A mean filter of 1 pixel was introduced, followed by tubeness with a sigma factor of 0.05. Thresholding with RenyiEntropy was applied, and the image was converted into a mask with subsequent fill holes and watershed operations. Signals from both masks were combined, and watershed was applied. The analyze particles option identified particles with sizes over 25 pixels. Finally, manual removal of artifacts was performed. Cell area sizes were measured manually. The sum of the cortical ER area was divided by the area of the corresponding cell.

## Time-lapse microscopy

In all, 1.5% agarose pads were prepared by mixing agarose (Sigma A9539-500G) in media containing 2% glucose and 0.2 mM CuSO₄. A 50-µl drop was placed in middle of dimpled glass slide and promptly flattened. Yeast cells were placed on the pad and covered with a 22 × 40 mm coverslip. Time-lapse images were recorded using a Zeiss AxioObserver Z1 inverted microscope, equipped with a Plan Apochromat 100×1.4 N/A Oil immersion lens and a large chip sCMOS mono camera for sensitive fluorescence imaging (ORCA Flash 4 v2). Cells were placed in an incubated chamber set at 30 °C. Time lapses were recorded for up to two hours with 5 min intervals. For each time-point, 7–8 though focus series were acquired spaced 0.5 µm apart.

To adjust for photobleaching, we brought up signals by using a custom-made macro in ImageJ. For that we duplicated the image and run mean of 2 pixels, then subtracted background with 50 pixels rolling ball background subtraction, then subtracted 15 pixels and unsharped mask with a radius of 5 pixels and a mask weight of 0.7. Then we averaged the signal of the processed image with the

original image using image calculator. This process was repeated several times depending on signal quality.

## Imaris 3D reconstructions

To generate 3D reconstructions of the ER from Elyra images (Fig. 7), we utilized Imaris 10.1.0 software. To create a 3D representation of the ER network, we used the "Object" tool. Because of the complexity of the wild-type ER membrane network, we separated the wild-type cell into three segments: top, middle, and bottom. For the Lro1* reconstruction, patches of the fragmented ER were generated individually for greater accuracy. In both cases, the nucleus and artificially generated structures were manually removed.

## Radiolabeling and analysis of lipids

The steady-state labeling of lipids with [2-$^{14}$C]acetate was performed as described previously (Morlock et al, 1988). Lipids were extracted (Bligh and Dyer, 1959) from the radiolabeled cells, and then separated by one-dimensional TLC for neutral lipids (Henderson and Tocher, 1992). The resolved lipids were visualized by phosphorimaging and quantified by ImageQuant software using a standard curve of [2-$^{14}$C]acetate. The identity of radiolabeled lipids was confirmed by comparison with the migration of authentic standards visualized by staining with iodine vapor.

## Lipidomics analysis

The following chemicals were used throughout the lipid extraction and analysis, water (Hypergrade for LC-MS, LiChrosolv®, MerckKGaA), acetonitrile (MSsuprasolve®, Sigma Aldrich), methanol (HPLC grade ≥99.9%, Sigma Aldrich), chloroform (stabilsed with amylenes, ≥99.5%, Sigma Aldrich), isopropanol (Optima™LC/MS Grade, Fisher Scientific), ammonium formate (99.995%, Sigma Aldrich).

### Lipid extraction

Cells were pelleted, washed once with water, immediately snap frozen in liquid nitrogen and stored at −80 °C until processing. Yeast cells were removed from the freezer and placed on ice, 2–3 scoops (~100 mg) of glass beads were added to each tube, along with chloroform (650 µl) and methanol (350 µl). Internal standards (10 µl, SPLASH LpidoMIX, Avanti Polar Lipids) were added, lids were placed on vials, cells were fractured using a "bead beater" for 4 × 60 s, resting 60 s on ice between repeats. Samples were sonicated for 5 min at 37 °C, water (400 µl) was added and samples were vortexed (30–60 s each). After centrifugation (132,500 rpm, 5 min), the bottom organic layer was collected using a glass pipette, transferred to a borosilicate test tube and dried under N$_2$ at 37 °C. Samples were resuspended in chloroform (100 µl) and added to amber HPLC vials, fitted with insert vial, before drying at 37 °C under N$_2$. Before injection samples were resuspended by vortexing in 100 µl isopropanol: methanol (1: 1).

### Lipidomic data acquisition by high-performance liquid chromatography mass-spectrometry (HPLC- MS)

HPLC-MS analysis was by hybrid quadrupole Orbitrap mass spectrometer (Q Exactive, ThermoScientific) fitted with an ultra-high-performance LC system (Ultimate 3000, ThermoScientific). Separation was achieved using reversed-phase chromatography with an ACQUITY UPLC HSS T3 Column (100 Å, 1.8 µm, 2.1 mm × 100 mm, Waters Corporation), held at 40 °C in the column oven. The following solvent gradient was used, consisting of solvent A; (60:40 v/v); water: acetonitrile and 10 mM ammonium formate and solvent B; (90:10) isopropanol: acetonitrile and 10 mM ammonium formate. A flow rate of 200 µL/min was used throughout diverting flow to waste for the first 30 s. The initial gradient was 0% B (0–1 min), which rose to 100% B over 53 min, holding at 100% B for 3 min, before reducing back to 0% B in 1 min and equilibrating for 9 min. Samples were stored at 4 °C in the autosampler and 5 µL was injected using µL pick-up injection. Each sample was injected twice and analyzed by mass spectrometry in the positive and negative mode, respectively, using heated electrospray ionization (HESI). Full experimental and instrument settings have been published previously (Bahja et al, 2022). Lipid identification was performed using Lipidex (Hutchins et al, 2018) and MZmine (Katajamaa et al, 2006; Pluskal et al, 2010) using the settings reported previously (Bahja et al, 2022).

## Immunoblotting

Yeast cells (6 OD$_{600nm}$) were harvested and washed with sterile water. Cells were lysed in 100 µl SDS-sample buffer with 0.5 mm diameter glass beads (BioSpec Products, Bartlesville, OK) by two rounds of boiling for 2 min and vortexing for 30 s. Protein extracts were centrifuged at 13,000 rpm for 15 min, and the supernatants analyzed by western blot. Western blot signals were developed using ECL (GE Healthcare, Little Chalfont UK). Anti-Dgk1 antibody was described in (Han et al, 2008b); detection of Protein-A fusions was done by using horseradish peroxidase (HRP) conjugated goat anti-rabbit IgG (BD Biosciences, 554021).

## Molecular modeling

A dimer of Lro1 was created using the ColabFold implementation of AlphaFold2 (Jumper et al, 2021; Mirdita et al, 2022). This model predicted that the TM helix of Lro1 bound to the active site of the enzyme. As this would not permit the helix to span the membrane, we additionally modeled the TM helices as a dimer and positioned them below the soluble domain. Using this structure as a template, we applied Modeller v10 to build the wild-type Lro1 and Lro1* (Webb and Sali, 2016). In each case, the models were built from residue R71 to M661.

## Coarse-grained MD simulations

For the CG simulations, the protein was converted to the Martini 3 forcefield using martinize2 (https://doi.org/10.7554/eLife.90627.2) including a 1000 kJ mol$^{-1}$ nm$^{-2}$ elastic network between C-alpha atoms. The Lro1 dimer was then embedded into a POPC lipid membrane using insane (Wassenaar et al, 2015), which was followed by solvation with Martini water and neutralized with 150 M NaCl. Energy minimizations of the systems was performed using the steepest descents method in all cases. A timestep of 20 fs was used for production simulations, with five independent simulations of 1 µs for each Lro1 construct.

## Atomistic simulations

Atomistic coordinates were generated from the end snapshots of the CG simulations using CG2AT2 (Vickery and Stansfeld, 2021). Atomistic molecular simulations were performed with the CHARMM36m forcefield (Huang et al, 2017). A timestep of 2 fs was used for production simulations, with three independent simulations of 500 ns for Lro1 and Lro1*. For all simulations, the C-rescale barostat (Bernetti and Bussi, 2020) was set at 1 bar and the velocity-rescaling thermostat (Bussi et al, 2007) was used at 310 K. All simulations were performed using Gromacs 2022.3 (Abraham et al, 2015). The RMSFs of the proteins were measured using gmx tools. Molecular visualization was performed with PyMOL (The PyMOL Molecular Graphics System, Version 3.0 Schrödinger, LLC).

## Statistical analysis

Graphs and statistical analyses were performed using GraphPad Prism. Graphs are means with standard deviations. We conducted the Shapiro–Wilk test to establish data distribution. Once Gaussian distribution was confirmed, we checked the data variance using the Brown–Forsythe and Bartlett's tests; if both tests passed, we applied Student's *t* tests for comparisons involving two samples or one-way ANOVA for analyses involving more than two samples, followed by Šidák correction for multiple comparisons. When either Brown–Forsythe or Bartlett's test failed, we conducted Welch *t* test for comparisons involving two samples and Brown–Forsythe and Welch ANOVA tests for analyses involving more than two samples, followed by Dunnett's T3 correction for multiple comparisons. *P* values were coded as follows: $*P < 0.05$, $**P < 0.01$, $***P < 0.001$, and $****P < 0.0001$.

# Data availability

The raw mass spectrometry lipidomics data were deposited in Mendeley: Mendeley Data https://doi.org/10.17632/kkhz43nxwz.1 https://data.mendeley.com/datasets/kkhz43nxwz/1.

The source data of this paper are collected in the following database record: biostudies:S-SCDT-10_1038-S44318-024-00355-3.

# Peer review information

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

## Acknowledgements

The authors thank Paul Luzio and Helena Santos-Rosa for comments on the manuscript; Sebastian Schuck and Bernard Kelly for reagents; Matthew Gratian and Mark Bowen for help with microscopy; and Peter Sterk and Vasileios Kargas for help with bioinformatic and structural analysis. This work was supported by a Biotechnology and Biological Sciences Research Council (BBSRC) grant [BB/T005610/1] to SS; a Wellcome Trust equipment grant (212892/Z/18/Z) to the Cambridge Institute for Medical Research; a National Institutes of Health Grant GM136128 from the United States Public Health Service to GMC. The content is solely the responsibility of the authors and does not necessarily represent the official views of the National Institutes of Health. PJS acknowledges Wellcome (208361/Z/17/Z), BBSRC, MRC, and the Howard Dalton Centre for funding. PJS acknowledges Sulis at HPC Midlands+, which was funded by the EPSRC on grant EP/T022108/1, and the University of Warwick Scientific Computing Research Technology Platform for computational access. This project made use of time on ARCHER2 and JADE2 granted via the UK High-End Computing Consortium for Biomolecular Simulation, HECBioSim (http://www.hecbiosim.ac.uk), supported by EPSRC (grant no. EP/R029407/1).

## Author contributions

**Pawel K Lysyganicz**: Conceptualization; Data curation; Formal analysis; Validation; Investigation; Methodology; Writing—review and editing. **Antonio D Barbosa**: Resources; Formal analysis; Visualization; Writing—review and editing. **Shoily Khondker**: Formal analysis; Validation; Investigation; Writing—review and editing. **Nicolas A Stewart**: Resources; Methodology; Writing—review and editing. **George M Carman**: Formal analysis; Supervision; Funding acquisition; Methodology; Writing—review and editing. **Phillip J Stansfeld**: Formal analysis; Funding acquisition; Validation; Investigation; Methodology; Writing—review and editing. **Marcus K Dymond**: Resources; Formal analysis; Validation; Investigation; Methodology; Writing—review and editing. **Symeon Siniossoglou**: Conceptualization; Supervision; Funding acquisition; Validation; Investigation; Visualization; Writing—original draft; Project administration; Writing—review and editing.

Source data underlying figure panels in this paper may have individual authorship assigned. Where available, figure panel/source data authorship is listed in the following database record: biostudies:S-SCDT-10_1038-S44318-024-00355-3.

## Disclosure and competing interests statement

The authors declare no competing interests.

# Expanded View Figure

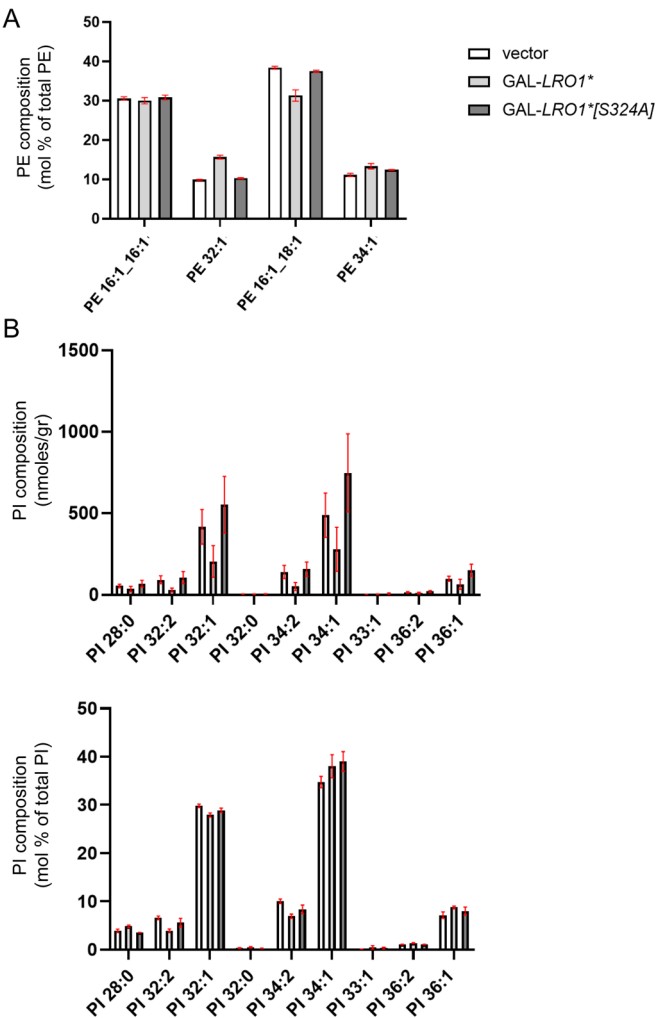

**Figure EV1.  Wild-type cells (BY4741) carrying the indicated constructs were grown in galactose as in Fig. 3A and processed for lipidomics analysis as described under Methods; data are means ± SD from three experiments.**

(A) Analysis of the major PE species. (B) Analysis of PI species; PI data are shown both as nmoles/gr (top panel) and mol% of total PI (bottom panel).

