## [Peer Review File · The EMBO Journal]

Partitioning of fatty acids between membrane and storage lipids controls ER membrane expansion

Pawel Lysyganicz, Antonio Barbosa, Shoily Khondker, Nicolas Stewart, George Carman, Phillip Stansfeld, Markus Dymond, and Symeon Siniossoglou

Corresponding author(s): Symeon Siniossoglou (ss560@cam.ac.uk)

Review Timeline:

Submission Date:	25th Jun 24
Editorial Decision:	9th Aug 24
Revision Received:	5th Nov 24
Editorial Decision:	26th Nov 24
Revision Received:	5th Dec 24
Accepted:	13th Dec 24

Editor: William Teale

Transaction Report:

Dear Dr. Siniosoglou,

Thank you again for the submission of your manuscript entitled "The partitioning of fatty acids between membrane and storage lipids controls ER membrane expansion" and for your patience during this unusually long review process. We have now received the reports from the referees, which I copy below.

As you can see from their comments, while referee #2 requests a deeper mechanistic exploration of Lro1 regulation, all referees state that your work is of interest and well done. That said, all of them raise concerns that will require your attention before your manuscript can be published in The EMBO Journal. While providing all requested structural insights would take the work beyond the scope of this manuscript, some analysis of why lid domain removal activates Lro1 would be beneficial.

Based on the overall interest expressed in the reports, however, I would like to invite you to address the comments of all referees in a revised version of the manuscript. I should add that it is The EMBO Journal policy to allow only a single major round of revision and that it is therefore important to resolve the main concerns at this stage. I believe the concerns of the referees are reasonable and addressable, but please contact me if you have any questions, need further input on the referee comments or if you anticipate any problems in addressing any of their points. I am always available to discuss these reports over Zoom; getting editorial input on a revision plan before extra experiments are initiated often saves time and resources. Please, follow the instructions below when preparing your manuscript for resubmission.

I would also like to point out that as a matter of policy, competing manuscripts published during this period will not be taken into consideration in our assessment of the novelty presented by your study ("scooping" protection). We have extended this 'scooping protection policy' beyond the usual 3 month revision timeline to cover the period required for a full revision to address the essential experimental issues. Please contact me if you see a paper with related content published elsewhere to discuss the appropriate course of action.

Again, please contact me at any time during revision if you need any help or have further questions.

Thank you very much again for the opportunity to consider your work for publication. I look forward to your revision.

Best regards,

William

William Teale, Ph.D.
Editor
The EMBO Journal

When submitting your revised manuscript, please carefully review the instructions below and include the following items:

- 1) a .docx formatted version of the manuscript text (including legends for main figures, EV figures and tables). Please make sure that the changes are highlighted to be clearly visible.
- 2) individual production quality figure files as .eps, .tif, .jpg (one file per figure).
- 3) a .docx formatted letter INCLUDING the reviewers' reports and your detailed point-by-point response to their comments. As part of the EMBO Press transparent editorial process, the point-by-point response is part of the Review Process File (RPF), which will be published alongside your paper.
- 4) a complete author checklist, which you can download from our author guidelines ([https://wol-prod-cdn.literatumonline.com/pb-assets/embo-site/Author Checklist%20-%20EMBO%20J-1561436015657.xlsx](https://wol-prod-cdn.literatumonline.com/pb-assets/embo-site/Author%20Checklist%20-%20EMBO%20J-1561436015657.xlsx)). Please insert information in the checklist that is also reflected in the manuscript. The completed author checklist will also be part of the RPF.
- 5) Please note that all corresponding authors are required to supply an ORCID ID for their name upon submission of a revised manuscript.
- 6) We require a 'Data Availability' section after the Materials and Methods. Before submitting your revision, primary datasets

produced in this study need to be deposited in an appropriate public database, and the accession numbers and database listed under 'Data Availability'. Please remember to provide a reviewer password if the datasets are not yet public (see <https://www.embopress.org/page/journal/14602075/authorguide#datadeposition>). If no data deposition in external databases is needed for this paper, please then state in this section: This study includes no data deposited in external repositories. Note that the Data Availability Section is restricted to new primary data that are part of this study.

Note - All links should resolve to a page where the data can be accessed.

8) For data quantification: please specify the name of the statistical test used to generate error bars and P values, the number (n) of independent experiments (specify technical or biological replicates) underlying each data point and the test used to calculate p-values in each figure legend. The figure legends should contain a basic description of n, P and the test applied. Graphs must include a description of the bars and the error bars (s.d., s.e.m.).

9) We would also encourage you to include the source data for figure panels that show essential data. Numerical data can be provided as individual .xls or .csv files (including a tab describing the data). For 'blots' or microscopy, uncropped images should be submitted (using a zip archive or a single pdf per main figure if multiple images need to be supplied for one panel). Additional information on source data and instruction on how to label the files are available at .

10) We replaced Supplementary Information with Expanded View (EV) Figures and Tables that are collapsible/expandable online (see examples in <https://www.embopress.org/doi/10.15252/embj.201695874>). A maximum of 5 EV Figures can be typeset. EV Figures should be cited as 'Figure EV1, Figure EV2" etc. in the text and their respective legends should be included in the main text after the legends of regular figures.

12) Our journal encourages inclusion of *data citations in the reference list* to directly cite datasets that were re-used and obtained from public databases. Data citations in the article text are distinct from normal bibliographical citations and should directly link to the database records from which the data can be accessed. In the main text, data citations are formatted as follows: "Data ref: Smith et al, 2001" or "Data ref: NCBI Sequence Read Archive PRJNA342805, 2017". In the Reference list, data citations must be labeled with "[DATASET]". A data reference must provide the database name, accession number/identifiers and a resolvable link to the landing page from which the data can be accessed at the end of the reference. Further instructions are available at .

13) In order to increase the reproducibility and reach of your work, The EMBO Journal includes a table of reagents that were used in the study. Please provide this along with your revisions.

Further instructions for preparing your revised manuscript:

When assembling figures, please refer to our figure preparation guideline in order to ensure proper formatting and readability in

print as well as on screen:

We realize that it is difficult to revise to a specific deadline. In the interest of protecting the conceptual advance provided by the work, we recommend a revision within 3 months (7th Nov 2024). Please discuss the revision progress ahead of this time with the editor if you require more time to complete the revisions. Use the link below to submit your revision:

Referee #1:

This study examines the role of Phospholipid Acyltransferases (PDATs) in regulating ER membrane homeostasis by transferring fatty acids (FAs) from membrane phospholipids into lipid droplets. PDATs are atypical diacylglycerol acyltransferases that utilize organelle membranes as a FA source to generate triglycerides. Unlike Diacylglycerol Acyltransferases (DGATs), whose roles in FA metabolism are well understood, PDATs' biological functions have been elusive. In this study, Lysyganicz and colleagues demonstrate that PDAT activity mediate the retraction of ER expansion by promoting the transfer of FAs from membrane phospholipids into lipid droplets. They also show that the conserved phosphatidic acid phosphatase Pah1 supplies the diacylglycerol pool required for PDAT-mediated ER membrane breakdown, independent of autophagy. Using an engineered version of the yeast PDAT, Lro1*, which lacks an inhibitory domain, they show that enhanced PDAT activity is due to direct access to ER membrane phospholipids and DG. Their findings suggest that Pah1 and PDAT activities are coordinated to inhibit ER membrane biogenesis, particularly at the nucleolar-associated membrane subdomain, and that PDAT-mediated ER membrane remodeling is independent of autophagy. Additionally, they identify the lysophospholipid acyltransferase Ale1 as crucial for re-acylating lysoPC products of Lro1*. This study establishes PDATs as novel regulators of organelle dynamics, highlighting their role in maintaining membrane homeostasis and suggesting that their primary function may be organelle remodeling rather than energy storage. The presence of similar pathways in other organisms, including humans, underscores the evolutionary conservation of PDAT-mediated lipid regulation.

This study integrates a wide array of genetic, biochemical, imaging, and in silico approaches, all supported by appropriate controls. The manuscript is remarkably well-written, and the images and quantifications are thorough and detailed. I firmly believe that this wonderful study deserves publication in The EMBO Journal with very minimal revision. Here are my minor comments:

- To gain deeper mechanistic insights, it would be interesting to explore the conformational changes in Lro1 that allow the displacement of the inhibitory "lid" domain. Do the authors think that this conformational change might be regulated by the lipid composition of the target membrane? For example, could it be induced by increased DG levels? It might be interesting to simulate this using molecular dynamics. Or maybe the conformational changes could be due to Lro1 oligomerization and/or interaction with key protein regulators? Additionally, investigating the specific structural changes in wild-type Lro1 using cryo-EM or NMR spectroscopy, although perhaps beyond the scope of this manuscript, would be a valuable direction for future research.

- In Figure 6D, it is indicated that only about 20% of cells overexpressing DGK1 have normal ER morphology (associated with larger LDs). However, in Figure 6E, it appears that this percentage is higher (around 50%). The authors should clarify this discrepancy or replace the image with a more representative one.

Referee #2:

In this manuscript, the authors study the function and regulation of the yeast phospholipid diacylglycerol acyltransferase (PDAT) Lro1, which catalyzes the synthesis of triglycerides from DAG and phospholipids, i.e. conversion of membrane lipids into neutral lipids that are stored in lipid droplets. The study builds on previous work by these authors on the function of this enzyme, and takes advantage of an interesting new hyperactive mutant termed Lro1*. Taking advantage of this mutant, they explore the functional relationship between Lro1 and the PA phosphatase Pah1, which generates TAG, and the regulation of the ER membrane homeostasis.

The study is generally carefully done and presents several interesting observations, but it could benefit from some additional mechanistic insight. It is also not always easy to reconcile the results of this study with previous work.

Specific comments:

1. The Lro1* mutant is intriguing and represents a major part of this study, raising very interesting questions about the regulation of Lro1 activity. The authors suggest that this mutant is hyperactive because it lacks a "lid-like domain" on the ER luminal side of Lro1, composed of two membrane proximal helices, which controls the access of substrates (PL and DAG) to the active site of Lro1. This would imply that the position of these helices is regulated in some way, but this question is not explored, and the statement is solely based on the AlphaFold structural model of Lro1. This question would warrant some further exploration, or the first three subtitles of the results sections should be toned down.

- The first subsection, "Identification of a luminal regulatory "lid"-like domain in Lro1.", just represents the AlphaFold model, mostly focusing on Lro1 dimerization; there is no evidence of regulation of a lid-like domain. Are these helices conserved among Lro1 orthologues, and what specific features are conserved? What is the evidence that this region (Ser104 - Asp152) is an actual domain (ie., an independently folding subunit of the protein)?

- The cellular phenotypes of the Lro1* mutant, which lacks this region, are very interesting, but the exact molecular mechanism of Lro1* hyper-activity could be further explored. The GS string mutant suggests that positioning of the luminal domain with respect to the membrane, but it doesn't address how exactly the membrane proximal helices function. Is it possible to make some point mutants to disrupt specific features of these helices? Along the same lines, have the authors tried to purify and structurally characterize the luminal domain of Lro1, or test its interaction with membranes or its enzymatic activity in vitro?

- MD simulations in Fig. 4 suggest that Lro1* interacts more strongly with the membrane; however, this part of the study is not very well described or presented, and is therefore not very convincing. A more realistic bilayer, and including DAG, could also be used. It also should be noted that the MD simulation results depend on the accuracy of the Lro1/Lro1* structural models. At present, there is not sufficient data for subtitle 'The "lid"-like domain controls lipid accessibility to the catalytic site of Lro1'.

2. Localization of Lro1* is interesting and I would recommend to include these data in the main figure (Fig. EV6), and also to show the data on the localization of H1-Lro1*. In their previous work (Barbosa et al. 2019), the authors showed that the activity of Lro1 strongly depended on its localization to the inner nuclear membrane, and they demonstrated that the N-ter domain of Lro1 was necessary and sufficient for its targeting to a subdomain of the INM, which was important for maintaining nuclear integrity. This does not seem to be the case for Lro1*; the authors conclude that its elevated activity is independent of its cellular location. How do the authors reconcile the results from these two studies?

3. The lipidomics data in Fig. 3 shows a striking effect of Lro1* over-expression on total cellular PI levels. This is very interesting, but is not further explored or commented on in the discussion (and conflicts with the scheme in Fig. 7A). There is also a strong effect on the exact species of PC that are depleted or enriched by Lro1* O/E, even though the total levels of PC are not effected.

- One approach that might shed some light on these observations could be to perform a genome-wide screen with the Cu-inducible expression system of Lro1*, i.e., expand the targeted approach in Fig. 5, which was limited to only a few candidates (and the authors have previously shown that there was a strong connection between Lro1 and Ale1, so this result was probably not so surprising).

4. In all figures, the data is quantified and the observed effects from several independent experiments are large and convincing. However, the results of the statistical tests are not always convincing and are sometimes even distracting (see for example Fig. 6D or 6H). What was the sample size used for the statistical tests? It is not stated how many cells were quantified in each experiment (please include this information in the figure legends). I assume that the statistical tests were performed on the

results of individual experiments, otherwise the N is too small (performing a statistical tests using means of three experiments makes no sense).

Referee #3:

General Summary: This study by Lysyganicz et al. investigates the role of phospholipid diacylglycerol acyltransferases (PDATs) in regulating lipid synthesis in the endoplasmic reticulum (ER) membrane in yeast cells. PDATs, specifically the yeast enzyme Lro1, typically use membrane phospholipids as a source of fatty acids to produce triglycerides, which are then stored in lipid droplets. A hyperactive version of Lro1, called Lro1*, was engineered based on an AF prediction and membrane modeling to study this process. They found that Lro1 activity is regulated by a luminal domain that controls the enzyme's access to phospholipids from the ER membrane. Furthermore, the study demonstrates that, Lro1 and Pah1 (a member of the lipin family that converts PA to DG), can remove excess ER membrane thus controlling the ER size and this process is independent of autophagy, suggesting a novel mechanism for organelle membrane turnover regulated by the coordinated action of lipid enzymes. These findings highlight the importance of PDATs as regulators of organelle dynamics. Overall, this study is interesting and novel, and the manuscript is well-prepared. The results are sufficiently interpreted with few exceptions.

See the comments below:

- Figure 2E, plating assays are good to determine the sensitivity of a mutant strain to certain conditions compared to wildtype. To assess how growth is affected, growth assays i.e. measure the OD or cell concentration over time is required.
- Same comment for Figure 3C, 5B. To be able to conclude that growth is not significantly affected, growth curves need to be done.
- Figure 2E, H1-Lro1 and Lro1 mCH rescue lipotoxicity less efficiently than the other mutants. Can the authors comment on this?
- The authors mentioned simulations with "GS string" Lro1 but the figure is not included.
- The GS string Lro1 clearly rescues toxicity at 1x oleate. What is the explanation for this?
- The microscopy corresponding to Figure 4C should be included.
- Does the addition of choline reverse LD accumulation in Lro1* expressing cells?

Minor comments:

- In the abstract 2nd sentence, remove redundancy "remains" or "still": "How these pathways regulate organelle size, remains still poorly understood."
- Figures 1C, 4A, 4D require labeling of ER lumen and cytoplasm sides

RESPONSE TO REVIEWERS

We thank the reviewers for the time they took to carefully assess our work and for their constructive comments that gave us the opportunity to improve our manuscript. The revised paper contains additional data and clarifications that address the reviewers' comments. Please find below a point-by-point response to all the issues raised by the reviewers.

Sincerely and on behalf of all my collaborators and co-authors,

Symeon Siniosoglou

Reviewer 1

...This study integrates a wide array of genetic, biochemical, imaging, and in silico approaches, all supported by appropriate controls. The manuscript is remarkably well-written, and the images and quantifications are thorough and detailed. I firmly believe that this wonderful study deserves publication in The EMBO Journal with very minimal revision.

Here are my minor comments:

- To gain deeper mechanistic insights, it would be interesting to explore the conformational changes in Lro1 that allow the displacement of the inhibitory "lid" domain. Do the authors think that this conformational change might be regulated by the lipid composition of the target membrane? For example, could it be induced by increased DG levels? It might be interesting to simulate this using molecular dynamics. Or maybe the conformational changes could be due to Lro1 oligomerization and/or interaction with key protein regulators? Additionally, investigating the specific structural changes in wild-type Lro1 using cryo-EM or NMR spectroscopy, although perhaps beyond the scope of this manuscript, would be a valuable direction for future research.

> We thank the reviewer for their positive assessment of our study; we have indeed considered the possible role of DG at the conformation of the lid at length but testing this hypothesis using molecular simulations is challenging: the main issue is that DG as a lipid is almost featureless, so interactions are not driven by charge, making the correct starting configuration for the simulations almost impossible. With respect to Lro1 protein interactors, we have now identified a number of candidates from both the luminal and cytosolic side but validating them and testing their roles on lid dynamics will require time. We feel that exploring these questions is beyond the scope of this work, however we do mention in Discussion the three possible mechanisms (DG, oligomerization and protein interactors) that could regulate the lid. We thank the reviewer for suggesting to investigate the structural changes in Lro1 using cryo-EM, which is indeed an approach we aim to take.

- In Figure 6D, it is indicated that only about 20% of cells overexpressing DGK1 have normal ER morphology (associated with larger LDs). However, in Figure 6E, it appears that this percentage is higher (around 50%). The authors should clarify this discrepancy or replace the image with a more representative one.

> A more representative micrograph is now shown in Figure 6E.

Reviewer 2

..The study is generally carefully done and presents several interesting observations, but it could benefit from some additional mechanistic insight. It is also not always easy to reconcile the results of this study with previous work.

Specific comments: 1. The Lro1* mutant is intriguing and represents a major part of this study, raising very interesting questions about the regulation of Lro1 activity. The authors suggest that this mutant is hyperactive because it lacks a "lid-like domain" on the ER luminal side of Lro1, composed of two membrane proximal helices, which controls the access of substrates (PL and DAG) to the active site of Lro1. This would imply that the position of these helices is regulated in some way, but this question is not explored, and the statement is solely based on the AlphaFold structural model of Lro1. This question would warrant some further exploration, or the first three subtitles of the results sections should be toned down.

> We thank the reviewer for finding that our study presents several interesting observations. We provide below additional data and clarifications to address their comments. Exploring the detailed mechanism(s) by which the lid regulates the activity of Lro1 is currently under way but we feel that the relevant approaches represent separate lines of investigation which would take the work beyond the scope of this manuscript. We have accordingly toned down the subtitles and some of our statements within the three first sections of the results and within the discussion.

- The first subsection, "Identification of a luminal regulatory "lid"-like domain in Lro1.", just represents the AlphaFold model, mostly focusing on Lro1 dimerization; there is no evidence of regulation of a lid-like domain. Are these helices conserved among Lro1 orthologues, and what specific features are conserved? What is the evidence that this region (Ser104 - Asp152) is an actual domain (ie., an independently folding subunit of the protein)?

> We have now removed the term "regulatory" and titled this section as "Modelling of the protein structure of Lro1". We admit that we used the term "domain" with its broader sense rather than with its strict structural definition - there is no experimental evidence for Ser104 - Asp152 folding and acting independently. We no longer refer to this sequence as "domain" or "lid" although we kept the latter term for the discussion. To address the question on the conservation of the helices, we compared the two largest taxonomic groups of PDATs - fungi and plants; although the enzymes have overall comparable amino acid residue lengths, we found that in plants the luminal "linker" connecting the transmembrane and catalytic domains is considerably shorter. This is more apparent in MSA of fungal and plant PDATs showing that in the latter, a large part of the linker is missing. We have now included these data at the beginning of the results and in Appendix Figure S1. Although this could suggest a distinct regulation in fungal enzymes via the "lid", please note that AlphaFold models of plant PDATs predict a single shorter helix parallel to the membrane; exploring these differences further will require a more detailed structural genomics approach.

- The cellular phenotypes of the Lro1* mutant, which lacks this region, are very interesting, but the exact molecular mechanism of Lro1* hyper-activity could be further explored. The GS string mutant suggests that positioning of the luminal domain with respect to the membrane, but it doesn't address how exactly the membrane proximal helices function. Is it possible to make some point mutants to disrupt specific features of these helices? Along the same lines, have the authors tried to purify and structurally characterize the luminal domain of Lro1, or test its interaction with membranes or its enzymatic activity in vitro?

> We have identified a predicted amphipathic helix (AH) within the lid domain (Fig. Rev1A and 1B); mutating the residues within the hydrophobic phase of the AH to disrupt its amphipathic nature, decreases the in vivo activity of Lro1 (Fig. Rev1C and 1D). We therefore hypothesize that these residues participate in membrane binding and/or substrate accessibility. Additionally, the prediction shows two interfacial lysines (in blue in Fig. Rev1A) which could be involved in interactions with phospholipid headgroups. However, we feel that these data - shown below for the reviewer's interest - are of preliminary nature and do not add much to the manuscript

so we decided to not include them in the revised manuscript.

Previous work has shown that the luminal part of Lro1 (residues 98 – 661) is not active in vivo (Choudhary et al, JCB 2020), so we have not attempted to express it in vitro. We are currently attempting to reconstitute purified full-length Lro1 from yeast in membrane carriers for activity assays and structural analysis; these experiments are not easy and we feel they represent an entire project outside the scope of the present study.

- MD simulations in Fig. 4 suggest that Lro1* interacts more strongly with the membrane; however, this part of the study is not very well described or presented, and is therefore not very convincing. A more realistic bilayer, and including DAG, could also be used. It also should be noted that the MD simulation results depend on the accuracy of the Lro1/Lro1* structural models. At present, there is not sufficient data for subtitle 'The "lid"-like domain controls lipid accessibility to the catalytic site of Lro1'.

> We agree that investigating the effect of DG in the Lro1 simulations would be interesting and have considered this issue thoroughly. However, in the nanosecond timescale simulated here, testing the role of DG using MD simulations is challenging. As detailed to reviewer 1, because DG as a lipid is almost featureless, interactions are not driven by charge and therefore getting the correct starting configuration for the MD simulations is almost impossible. Instead, we have now modelled the binding of DG to wild-type Lro1 and predict it to bind at the residues of the active site (new Appendix Fig. S7C and D); this is consistent with the results of the MD simulations showing that lipid extraction in Lro1 takes place towards the active site of the*

enzyme. These data are now included in the results section describing the simulations.

We agree with the reviewer that the simulations depend on the protein models used. To address this comment, and further validate our results, we now show that during atomistic MD simulations, the lid segment does retain its secondary structure and is also relatively stable; this suggests that despite the lower confidence from AlphaFold on the structure of the lid, the physical properties of the models are sound. These data are now shown in Appendix Fig. S7A and B. We have modified the text of this section to clarify these conclusions and changed the subtitle accordingly. We also now state in the discussion that in the absence of an experimentally verified structure of Lro1, we cannot rule out that removal of the predicted helical region induces the activity of Lro1 through alternative mechanisms.

2. Localization of Lro1* is interesting and I would recommend to include these data in the main figure (Fig. EV6), and also to show the data on the localization of H1-Lro1*. In their previous work (Barbosa et al. 2019), the authors showed that the activity of Lro1 strongly depended on its localization to the inner nuclear membrane, and they demonstrated that the N-ter domain of Lro1 was necessary and sufficient for its targeting to a subdomain of the INM, which was important for maintaining nuclear integrity. This does not seem to be the case for Lro1*; the authors conclude that its elevated activity is independent of its cellular location. How do the authors reconcile the results from these two studies?

> We now show the localization of H1-Lro1, together with Lro1 and Lro1*, in Fig. 2F. In Barbosa et al (2019), we showed that Lro1 is active in the quadruple acyltransferase (4D) mutant in stationary phase, when the enzyme localizes to the inner nuclear membrane. Our data in the Barbosa paper did not address the activity of Lro1 at the ER. We did not conclude at any point that Lro1 is not active when found at the ER at the exponential phase, or that Lro1 is active exclusively at the inner nuclear membrane. In our current study we indeed suggest that the elevated activity of Lro1* is independent of the membrane compartment it resides. The reason behind Lro1*'s localization to the ER, as opposed to the dual ER-inner nuclear membrane localization of wild-type Lro1, is not known. This could be due to a structural effect – for example shortening the luminal linker that connects the catalytic domain to its transmembrane domain may impede the passage of Lro1 through the nuclear pore membrane; alternatively, we show that Pah1 activity increases the targeting of Lro1* to the INM, suggesting that DG levels may be important for INM localization (Fig. 8); therefore, the decrease in DG levels in Lro1*-expressing cells (Figs. 2D and 3D) could affect the retention of Lro1* to the INM subdomain. Either possibility does not contradict the key role that the N-terminal domain plays in targeting Lro1 to the INM in a wild-type context. To clarify these issues, we are now discussing the localization of Lro1* in page 9 (Discussion).*

3. The lipidomics data in Fig. 3 shows a striking effect of Lro1* over-expression on total cellular PI levels. This is very interesting, but is not further explored or commented on in the discussion (and conflicts with the scheme in Fig. 7A). There is also a strong effect on the exact species of PC that are depleted or enriched by Lro1* O/E, even though the total levels of PC are not effected.

> The effects on PI are indeed striking as, unlike PC and PE, they are detected to all species; we envision two possibilities: (a) PI could be a direct substrate of Lro1, although none of the previous studies in yeast found any evidence for this; to further look into this, we checked total lyso-PI levels and found that they increase although the changes are not statistically significant – please bear in mind that we used the PI standard to quantify lyso-PI as we lacked a lyso-PI standard; addressing this*

possibility directly would require a PDAT activity assay with purified Lro1; (b) alternatively, the effects on PI may be indirect: the partitioning of CDP-DG for the synthesis of PI and the PS-generated PE/PC, which is governed by the enzymes Pis1 and Cho1 respectively, may be altered in Lro1*-overexpressing cells in order to maintain PE and PC levels. We now comment on these possibilities in Discussion. We also added the PI in the schemes shown in Figs. 1A and 6A. With respect to the species-specific effects of Lro1* on PC and PE vs their total levels, we did comment on this on both the results section (page 4) and the discussion (page 9).*

- One approach that might shed some light on these observations could be to perform a genome-wide screen with the Cu-inducible expression system of Lro1*, i.e., expand the targeted approach in Fig. 5, which was limited to only a few candidates (and the authors have previously shown that there was a strong connection between Lro1 and Ale1, so this result was probably not so surprising).

> Thank you for this suggestion; this screen is actually under way currently for both positive and negative Lro1-growth modifiers; we feel that the validation and characterization of hits from such a screen is a major engagement which falls outside the scope of the study and is more appropriate for future work.*

4. In all figures, the data is quantified and the observed effects from several independent experiments are large and convincing. However, the results of the statistical tests are not always convincing and are sometimes even distracting (see for example Fig. 6D or 6H). What was the sample size used for the statistical tests? It is not stated how many cells were quantified in each experiment (please include this information in the figure legends). I assume that the statistical tests were performed on the results of individual experiments, otherwise the N is too small (performing a statistical tests using means of three experiments makes no sense).

> We apologize for this omission (i.e. cell numbers); we now describe within each figure legend, the number of independent experiments performed, the minimum number of cells quantified per strain and per experiment, the statistical test and correction used, and p values. In the revised statistical methods we now describe the tests we used to establish data distribution and variance. We are not sure we understand what the reviewer means by “performing a statistical test using means of three experiments makes no sense”; in the study we are performing statistical analysis using means from three to seven independent groups, i.e. biological repeats, each comprising values from hundreds of cells.

Reviewer 3

..These findings highlight the importance of PDATs as regulators of organelle dynamics. Overall, this study is interesting and novel, and the manuscript is well-prepared. The results are sufficiently interpreted with few exceptions. See the comments below:

• Figure 2E, plating assays are good to determine the sensitivity of a mutant strain to certain conditions compared to wildtype. To assess how growth is affected, growth assays i.e. measure the OD or cell concentration over time is required.

> We thank the reviewer for their positive assessment of our study; we have now performed growth curves of the quadruple mutant (4D) expressing vector, Lro1 or Lro1 in the presence of oleate. We find a small but significant decrease in the generation time of Lro1*. These data are shown in the new Appendix Fig. S6A. We have removed from Figure 2E the spot assays of mCherry-tagged Lro1 and Lro1* strains as these were redundant with the LD Fig. 2C of the same strains and the*

original Fig. 2E was very crowded.

- Same comment for Figure 3C, 5B. To be able to conclude that growth is not significantly affected, growth curves need to be done.

> We now have performed growth curves of the strains shown in Fig. 3C, grown in galactose-containing media. The data confirm that overexpression of Lro1 strongly inhibits growth in a catalytic activity-dependent manner. These data are now shown in Appendix Fig. S6B. The catalytically dead Lro1* delays growth to a lesser degree, as does that of Lro1 (a fact that is also apparent at the dot spots in Fig. 3C), suggesting some additional non-catalytic effects of Lro1* and Lro1 overexpression. With respect to Fig. 5B, we have now re-phrased the relevant part of the text, stating that these mutations “did not stop the growth of Lro1*-expressing cells”.*

- Figure 2E, H1-Lro1 and Lro1 mCh rescue lipotoxicity less efficiently than the other mutants. Can the authors comment on this?

> H1-Lro1 is indeed less efficient in rescuing lipotoxicity of 4D, as also shown in Barbosa et al (2019). This phenotype of H1-Lro1 may be due to its reduced stability at the INM – protein levels of H1-Lro1 are stabilized in an Asi ubiquitin ligase mutant (Barbosa et al, 2019) – or stress when cells are forced to produce LDs throughout the entire INM. The reviewer is also correct in that Lro1-mCh appears to rescue less well lipotoxicity when compared to Lro1 – although not to the same extent as H1-Lro1; we do not know why that is; it is possible that the mCherry tag affects full activity of Lro1 when cells are challenged with high concentrations of oleate. However, please note that (a) in normal media, Lro1-mCh induces LD accumulation to a similar extent as Lro1* (Figs 2C vs Appendix S3); and (b) throughout our study we always compare Lro1 vs Lro1* or Lro1-mCh vs Lro1*-mCh.*

- The authors mentioned simulations with "GS string" Lro1 but the figure is not included.

> We apologize for this error; we had previously decided to not include the GS string simulations as we were not convinced on the quality of the model. We have now removed this mention from the text.

- The GS string Lro1 clearly rescues toxicity at 1x oleate. What is the explanation for this?

> With the GS string mutant, we altered the specific amino acid composition of the “linker” sequence, without affecting its length. The fact that this mutant does not phenocopy Lro1, which lacks altogether the linker, suggests that in Lro1* the overall positioning – possibly the increased proximity of the catalytic domain to the phospholipid bilayer - is important for elevating PDAT activity. However, in the GS string, the catalytic domain remains still attached to the transmembrane domain through the flexible Gly-Ser sequence which we postulate is enough to maintain its association with its lipid substrate and its enzymatic activity. This is consistent with the fact that the GS string maintains LD levels of the 4D strain at stationary phase.*

- The microscopy corresponding to Figure 4C should be included.

> We are now including the requested microscopy (Fig. 4C).

- Does the addition of choline reverse LD accumulation in Lro1* expressing cells?

> In response to this question, we quantified LD levels (BODIPY labelling) in wild-type cells expressing Lro1 or Lro1 under the control of the LRO1 promoter. We*

found that choline induces a small but significant decrease in LD levels in the Lro1-expressing cells, however these cells still accumulated significantly more LDs when compared to Lro1-expressing cells. These data are now shown in Appendix Fig. S8. In addition, choline supplementation does not reverse the growth inhibition of GAL-Lro1*- expressing cells (not shown). We hypothesize that, following the addition of choline, activation of the Kennedy pathway decreases some of the DG pool used by Lro1* for LD production, however this is not enough to reverse the accumulation of LDs and the growth inhibition.*

Minor comments:

- In the abstract 2nd sentence, remove redundancy "remains" or "still": "How these pathways regulate organelle size, remains still poorly understood."*

> Corrected.

- Figures 1C, 4A, 4D require labeling of ER lumen and cytoplasm sides*

> Labeling added.

Dear Symeon,

Thank you for submitting the revised version of your manuscript, which addresses the concerns of the referees. This revised version has now been re-reviewed; I attach the second referee report to the bottom of this mail. As you will see, you have addressed the referee's concerns satisfactorily. The referee asks that you include information on the robustness of MD simulations, which I agree would be a good idea; there are also some remaining editorial points which need to be addressed. In this regard, would you please:

- include up to five keywords,
- remove the author credit section from the manuscript file,
- include a callout for Figure 8C,
- add page numbers to the appendix table of contents,
- include a 'Reagents and Tools' table,
- save source data in a scheme of one figure per folder, uploading as .zip files; e.g. all the Source data files for figure 1 need to be saved in a single folder and this needs to be zipped and then uploaded as "SD figure 1.zip" file,
- provide specific URLs for Mendeley datasets are not provided in the data availability statement,
- define the annotated p values ** as well as provide the exact p-values for the same in the legend of figure 3D,
- state p values in the legends of figures 2C, 3A, B, D; 5C; 6D, G; 7D, G; 8C,
- define n, */ **/ ***/ ****, error bars and statistical test used n the legend of figure EV1 A-B,
- define white dotted lines in the legends of figures 2B and 7G,
- define red dotted line in the legends of figure 6F, and
- correct the section order to the following scheme: Title page - Abstract & Keywords - Introduction - Results - Discussion - Methods - Data Availability - Acknowledgements - Disclosure and Competing Interests Statement - References - Figure Legends - Table(s) - Expanded View Figure Legends.

We include a synopsis of the paper (see <http://emboj.embopress.org/>). Please provide me with a general summary image, two-sentence summary statement and 3-5 bullet points that capture the key findings of the paper.

I look forward to receiving these changes. EMBO Press is an editorially independent publishing platform for the development of EMBO scientific publications.

Best wishes,

William

William Teale, PhD
Editor
The EMBO Journal
w.teale@embojournal.org

We realize that it is difficult to revise to a specific deadline. In the interest of protecting the conceptual advance provided by the work, we recommend a revision within 3 months (24th Feb 2025). Please discuss the revision progress ahead of this time with the editor if you require more time to complete the revisions. Use the link below to submit your revision:

Referee #2:

The authors have addressed most of my comments in a satisfactory manner.

I still have some questions regarding the MD simulations presented in Fig. 4D & E. These simulations are crucial for the reasoning in the manuscript.

- The MD simulations in 4D/E were performed 5 times, however only single snap-shots are shown for WT and mutant. Were the lipid perturbations observed in all 5 simulations? Please provide some statistics on these observations, comparing Lro1 vs Lro1*, to show how reproducible and robust are these result.

- The figure legend states: "While Lro1 results in a planar lipid membrane, Lro1* permits the lipid headgroups to readily access the active site, resulting in perturbation of the membrane."

I don't see how this causality can be established. Furthermore, there are two active sites in the Lro1* dimer, but the perturbation is seen only on one side. Please rephrase this sentence.

In the future, it would be interesting to perform these simulations with a more realistic bilayer composition, as this could maybe shed some light on the observed differences in the abundance of specific lipid species.

Minor point:

The data and differences shown in Fig. 3D and EV1 are striking and convincing, but the asterisks aren't. I suggest removing them, like in other plots where N=3 (understandably for this type of experiment)

The EMBO Journal

<https://www.embopress.org/journal/14602075>

Wednesday the 4th of December, 2024

Re: EMBOJ-2024-118119 R1

Dear William,

Please find attached the revised version of our manuscript addressing the comments of reviewer 2 (below). We have incorporated the new data in the text, where we describe Figure 4D. We have also addressed all the editorial points raised and added the synopsis. I hope that our paper will be now suitable for publication in *The EMBO Journal*.

Sincerely and on behalf of all my collaborators and co-authors,

Symeon Siniossoglou

Response to reviewer 2:

I still have some questions regarding the MD simulations presented in Fig. 4D & E. These simulations are crucial for the reasoning in the manuscript. The MD simulations in 4D/E were performed 5 times, however only single snap-shots are shown for WT and mutant. Were the lipid perturbations observed in all 5 simulations? Please provide some statistics on these observations, comparing Lro1 vs Lro1*, to show how reproducible and robust are these result.

> Lipid deformations are observed in all five repeats of Lro1, illustrated for one repeat in Fig. 4D. Three repeats show consistent membrane deviations, while in two repeats this is only transient. We further assessed the proximity of the abstraction to the active sites of Lro1, by measuring the percentage of time a lipid is within 6 Å from the active site residue His618, as shown in Fig. 4E. In the three repeats that showed the greatest deformation, a lipid was in contact with His618 for 40.26 %, 37.86 % and 27.37 % of the simulation time. In the two other repeats this contact was far briefer at 0.50 % and 6.09 % of the simulated duration. In none of the Lro1 repeats was a lipid within 6 Å of the active site His618. We have now added this information in the results section (Figure 4D).*

The figure legend states: "While Lro1 results in a planar lipid membrane, Lro1* permits the lipid headgroups to readily access the active site, resulting in perturbation of the membrane." I don't see how this causality can be established. Furthermore, there are two active sites in the Lro1* dimer, but the perturbation is seen only on one side. Please rephrase this sentence.

> We have now replaced this sentence with the following: "In Lro1, the lipid bilayer remains planar while in Lro1 the lipid headgroups access the active site".*

In the future, it would be interesting to perform these simulations with a more realistic bilayer composition, as this could maybe shed some light on the observed differences in the abundance of specific lipid species.

> *This is an excellent suggestion for future work.*

Minor point: The data and differences shown in Fig. 3D and EV1 are striking and convincing, but the asterisks aren't. I suggest removing them, like in other plots where $N=3$ (understandably for this type of experiment)

> *We have now removed the asterisks from Fig. 3D and EV1.*

Dear Symeon,

I am pleased to inform you that your manuscript has been accepted for publication in the EMBO Journal.

Congratulations on a really interesting study!

Best wishes,

William

William Teale, PhD
Editor
The EMBO Journal
w.teale@embojournal.org
